# Comparative investigation of bagging enhanced machine learning for early detection of HCV infections using class imbalance technique with feature selection

Ekramul Haque Tusher[1], Mohd Arfian Ismail[1,2]*, Abdullah Akib[3], Lubna A. Gabralla[4], Ashraf Osman Ibrahim[5,6], Hafizan Mat Som[5,6], Muhammad Akmal Remli[7,8]

1 Faculty of Computing, Universiti Malaysia Pahang Al-Sultan Abdullah, Pahang, Malaysia, 2 Center of Excellence for Artificial Intelligence & Data Science, Universiti Malaysia Pahang Al-Sultan Abdullah, Lebuhraya Tun Razak, Gambang, Malaysia, 3 Industrial Engineering, Lamar University, Beaumont, Texas, United States of America, 4 Department of Computer Science, Applied College, Princess Nourah bint Abdulrahman University, Riyadh, Saudi Arabia, 5 Department of Computing, Universiti Teknologi PETRONAS, Seri Iskandar, Malaysia, 6 Positive Computing Research Center, Emerging & Digital Technologies Institute, Universiti Teknologi PETRONAS, Seri Iskandar, Malaysia, 7 Institute for Artificial Intelligence and Big Data, Universiti Malaysia Kelantan, Kota Bharu, Kelantan, Malaysia, 8 Faculty of Data Science and Computing, Universiti Malaysia Kelantan, Kota Bharu, Kelantan, Malaysia

* arfian@umpsa.edu.my

**Data availability statement:** Lichtinghagen KF, Hoffmann G. HCV data. UCI Machine Learning Repository. 1032 2020. DOI: https://doi.org/10.24432/C5D612.

## Abstract

Around 1.5 million new cases of Hepatitis C Virus (HCV) are diagnosed globally each year (World Health Organization, 2023). Consequently, there is a pressing need for early diagnostic methods for HCV. This study investigates the prognostic accuracy of several ensemble machine learning (ML) models for diagnosing HCV infection. The study utilizes a dataset comprising demographic information of 615 individuals suspected of having HCV infection. Additionally, the research employs oversampling and undersampling techniques to address class imbalances in the dataset and conducts feature reduction using the F-test in one-way analysis of variance. Ensemble ML methods, including Support Vector Machine (SVM), k-Nearest Neighbors (k-NN), Logistic Regression (LR), Random Forest (RF), Naïve Bayes (NB), and Decision Tree (DT), are used to predict HCV infection. The performance of these ensemble methods is evaluated using metrics such as accuracy, recall, precision, F1 score, G-mean, balanced accuracy, cross-validation (CV), area under the curve (AUC), standard deviation, and error rate. Compared with previous studies, the Bagging k-NN model demonstrated superior performance under oversampling conditions, achieving 98.37% accuracy, 98.23% CV score, 97.67% precision, 97.93% recall, 98.18% selectivity, 97.79% F1 score, 98.06% balanced accuracy, 98.05% G-mean, a 1.63% error rate, 0.98 AUC, and a standard deviation of 0.192. This study highlights the potential of ensemble ML approaches in improving the diagnosis of

**Funding:** This study was supported by Fundamental Research Grant (FRGS) with FRGS/1/2022/ICT02/UMP/02/2 (RDU220134) from the Ministry of Higher Education Malaysia and the authors extend their appreciation to the Deanship of Research and Graduate Studies at King Khalid University for funding this work through Large Research Project under grant number RGP2/319/45. The APC was funded by Nourah bint Abdulrahman University Researchers Supporting Project number (PNURSP2025R178), Princess Nourah bint Abdulrahman University, Riyadh, Saudi Arabia. The founders had no role in study design, data collection and analysis, decision to publish, or preparation of the manuscript.

**Competing interests:** The authors declare that they have no known competing financial interests or personal relationships that could have appeared to influence the work reported in this paper.

HCV. The findings provide a foundation for developing accurate predictive methods for HCV diagnosis.

## 1 Introduction

The HCV presents a challenge to global health. HCV functions as a pathogen, inducing hepatic inflammation that can result in various consequences, such as cirrhosis and hepatocellular carcinoma [1]. The genetic diversity of HCV is remarkable with seven genotypes and 67 subtypes [2]. This diversity is reflected in its distribution across all global regions, including North America, Europe, North Africa, Latin America, Southeast Asia, South Asia, Australia, West Africa. Middle East, Central Africa, and South Africa. Chronic HCV infection can be a lasting ordeal that gradually damages the liver over decades [3]. On the hand, acute HCV infections often go unnoticed due to their lack of symptoms. The scale of the threat posed by HCV becomes apparent when considering the numbers associated with it. Every year around 1.5 million people become affected by HCV according to the World Health Organization (WHO), it is believed that as 58 million individuals worldwide are currently dealing with chronic HCV infection [3].

Unfortunately, the HCV situation does not show signs of significant improvement. An estimation made in 2019 estimated that around 290,000 people died that year worldwide due to HCV infection, with a rate of 3.9 deaths per 100,000 individuals in 2019 [2]. These alarming death rates warns the world of the health consequences associated with HCV infection, and its long-term implications such as hepatocellular carcinoma and cirrhosis. Therefore, it is crucial to emphasize the need for strategies to diagnose early and treat HCV infection. The primary way HCV spreads are through contact with blood or blood products [1]. This transmission can occur in situations like substandard healthcare practices unsafe injections practices, unscreened blood transfusions or injection drug use. Additionally, HCV has the potential to be transmitted vertically from a mother to her offspring during the prenatal period or at the time of delivery, as well as through direct physical contact. Since there is no vaccine for HCV yet [2], taking precautionary measures becomes even more important. For instance, it is essential to provide care and screening for women while raising awareness about HCV infection among healthcare workers and the general public. Promoting sex practices along with ensuring a blood supply and safe injection practices are also crucial steps to consider. Moreover, special attention should be given to prioritize patients who show symptoms of HCV infection early on. In the context of diagnosing HCV infection, the predominant methodology is the identification of antibodies or the detection of viral RNA within the bloodstream [2].

While antibody tests offer some insights into current infections, they have limitations in differentiating between chronic infections or confirming viral clearance. RNA tests provide information about active infections and viral load which can contribute to treatment strategies and monitoring. Evaluating liver function and taking biopsies of the liver seems to be in figure out how bad the damage is and decide what kind of medicine to use. The treatment of this deadly virus infection has seen much positive growth in last few years with the introduction of direct-acting antiviral (DAA) drugs that address individual phases in the progression of this virus [4]. Despite advancements in HCV treatment with acting antiviral (DAA) drugs, limited access to diagnosis and treatment remains due to high costs, limited availability and lack of supply [2]. Therefore, there is avid need for more effective, fast and cost-efficient HCV early diagnostic techniques. Given the challenges and limitations of the current diagnosis methods, there seems to be a strong emphasis on leveraging ML techniques such as

supervised learning and data mining for classifying HCV patients. The potential of using data science, specifically data mining and ML must be noted.

Data science now acts as a major contributor in healthcare, specifically in the development of decision augmentation tools that use classification algorithms of ML for quick and sensitive decision-making. Numerous researches have made tremendous use of these algorithms for disease detection and early detection. ML tools such as k-NN, DT, SVM, RF, NB, LR, AdaBoost Classifier (ADA), and Extra Tree Classifier (ETC) have established themselves to hold multifaceted applications in numerous grounds including disease detection. Data mining is a procedure for excavating meaningful insight from extensive otherwise redundant datasets and it is another such tool that can be successfully leveraged to surpass the limiting factors in HCV diagnosis [5]. It can reveal risk factors [6] and biomarkers for HCV infection [7] and even predict disease progression. These approaches and potentials of data mining hold tremendous promise in the context of early detection of HCV or any other disease. Data mining techniques can also work with both labeled [8] and unlabeled data [9]. So, data mining techniques have been used to classify patients of many diseases into categories like "cured", "infected" or "healthy" [10–12]. To further enhance predictive accuracy and model robustness, this study employs a bagging-based ensemble approach. Bagging (Bootstrap Aggregating) is chosen due to its ability to reduce variance, mitigate overfitting, and improve generalization, particularly in medical datasets where data variability can impact classification performance. By aggregating multiple base learners trained on different subsets of data, bagging enhances stability and ensures that the predictive model does not overly rely on any single weak classifier. These models leverage the diversity introduced by bagging to make more reliable and sensitive predictions, thereby improving early diagnosis and personalized treatment plans. The advanced methodologies of disease detection with data mining offer unprecedented advantages over contemporary techniques of HCV detection. They not only can provide insights that may have been overlooked by conventional methods but also produce more understandable results and predictions that shed light on the underlying factors contributing to HCV infection. Furthermore, these techniques enable personalized diagnosis and treatment for HCV, leading to improved outcomes and enhanced quality of life. As a step forward in that process, this paper puts forth an ensemble ML approach for the precise classification of suspected HCV infections in patients. By combining the strengths of ML-based data mining methodologies, particularly through the incorporation of bagging algorithms, demonstrates that data mining techniques can revolutionize the early diagnosis and management of HCV.

## Contributions of the study

The study presented in this paper is underlined by a series of novel contributions to the field, which underscores not only the robust approach of this work towards handling and analyzing data but also the methodological advancements that was implemented. The following are the key novel contributions that this study made:

- To elevate the performance of ML models, this research work innovatively integrated bootstrap aggregation (Bagging) with multiple ML models including k-NN, RF, LR, DT, SVM, NB, and more. This integration offered a significant enhancement in model reliability and prediction power.
- This study differed from the traditional Synthetic Minority Over-sampling Technique (SMOTE) that is used in dealing with class imbalance. Instead, the researchers here utilized Under-sampling and Over-sampling techniques to create a balanced dataset, an

approach that has shown to be effective in preserving the natural distribution of the data while addressing the class imbalance issue.

- TIn a departure from typical feature extraction techniques, this study employed the one-way ANOVA F statistic test to conduct feature selection and feature reduction of the dataset. This statistically robust method allowed for more precise and valid selection of features, leading to improved model performance.

- The researchers undertook a rigorous comparison of the works of other researchers in the same field. This not only helped to position this work within the existing research landscape but also allowed the researchers to draw valuable insights from their findings to further enrich their study.

- Significantly, the novel approaches of this study resulted in the highest known accuracy in HCV infection prediction compared to the outcomes achieved by other researchers in this field (with Bagging k-NN at 98.37%). This breakthrough elevates the standards in this area of research and sets a new benchmark for future work.

The structure of this paper is as follows: Sect 2 provides an overview of the existing literature, establishing the context for our study. Sect 3 outlines the comprehensive methodology employed in our research. In Sect 4, we present our results and offer a thorough discussion of our findings. Finally, Sect 5 concludes the paper, summarizing key points and discussing implications for future research.

## 2 Related works

The healthcare industry has incorporated data science, particularly ML, into several applications. Decision support systems (DSS) in healthcare are created through the utilization of ML classification algorithms. These algorithms are employed to facilitate and enhance the process of decision making in the healthcare domain [13,37,38]. Research shows that classification algorithms are frequently employed for disease diagnosis [10–12].

As high performing algorithms are being used for disease diagnosis, ML methods like k-NN, DT, SVM, RF, NB, LR, AdaBoost, and Extra Trees (ET) have become essential tools for analyzing complex health datasets. The k-NN technique stands out as one of the utilized methods in ML. The DT finds application in areas like character recognition and medical diagnosis. The SVM effectively separate datasets through nonlinear boundaries. The RF is a ML algorithm that falls under the category of learning. It is commonly utilized to tackle classification and regression problems. The NB also stands as a classification algorithm that is widely used in fields, largely due to its simplicity and practicality. The LR method is also utilized in both statistical analysis and ML applications. The ADA is a ML method that is commonly employed for addressing classification tasks. Finally, the Extra Tree Classifier (ETC) integrates predictions from Decision Trees (DTs) in order to create a batch ML algorithm. Numerous research studies have explored the implementation of the aforementioned ML algorithms in health data analysis. For instance, In their study, Chang et al. devised a decision-making framework that employed decision trees (DTs) and genetic algorithms to effectively classify and identify iris, liver illness, breast carcinoma, and contraceptive method data, by considering different scenarios. In comparison with studies from earlier researchers who used Neural Networks (NN) and SVM for the same purpose, Chang et al. demonstrated that Case Based Fuzzy DTs (CBFDT) yielded superior results [14]. Seera et al. on the other hand put forth a proposal for an intelligent system that comprised RF models, a Fuzzy Min Max neural network, and a Classification and Regression Tree (CART). The researchers conducted an investigation into the efficacy of the decision support tool in classifying medical

data and discovered that it played a significant role in facilitating medical diagnosis [15]. Additionally, Nilashi et al. conducted a study aiming to predict diseases within breast cancer related datasets. They employed the fuzzy rule justification technique to accomplish this. They also successfully developed predictive models by uncovering fuzzy rules and proved that such ML models can indeed play a significant part in boosting accuracy of disease diagnosis [16].

Early detection of HCV disease surely plays a role in maintaining health. Numerous studies in the literature have estimated the prevalence of HCV disease and its associated conditions. Gower et al. emphasized the importance of conducting further studies to accurately and more effectively assess the burden of HCV disease [17]. On the other hand, Konerman et al. did a comprehensive analysis of HCV disease with the aim of identifying patients who are at risk of acquiring cirrhosis [18]. Additionally, the study conducted by Hashem et al. aimed to assess the efficacy of ML techniques in predicting fibrosis. Specifically, the researchers explored the integration of serum biomarkers and clinical data to construct classification models. The accuracy ranged from 66.3% to 84.4% in predicting fibrosis among patients with HCV using ML approaches. Therefore, it has been demonstrated that employing ML techniques may successfully forecast the likelihood of liver fibrosis resulting from hepatitis C [19]. In another study, Abd El-Salam et al. conducted research aiming to diagnose the disease through data analysis and classification techniques using ML algorithms for early forecast of HCV infection based on checking on cirrhotic patients [20]. The study incorporated a cohort of 4962 individuals diagnosed with hepatitis C, who were selected from fifteen medical facilities in Egypt over a period spanning from 2006 to 2017. The findings of the study demonstrated a level of accuracy up to 68.9%. Moreover, Nandipati et al. put forth a study using a dataset of patients to identify which specific traits are important, in predicting HCV. Their findings showed that the k-NN algorithm achieved the accuracy (51.06%) in the multi class category while the RF algorithm achieved an accuracy of 54.56% in the binary class category [21].

Another important area of research is predicting patients' response to HCV treatment through data classification. Previous studies have utilized methods such as Associative Classification, ANN and DT to develop classification models for this purpose [22]. During the research work by ElHefnawi, ANN and DT models demonstrated accuracies of 76% and 80% respectively [23]. Moreover, the analysis conducted by Neukam et al. involved the examination of demographic factors. The data involved in this study was acquired from cohorts of HIV/HCV coinfected people in Germany who were subject to prospective monitoring at an infectious disease monitoring center. In the study population, the researchers used both binary logistic univariate and stepwise multivariate logistic regression analyses to show that the HCV viral genotype, rs12979860 genotype, and baseline HCV RNA load were all linked to SVR (sustained virologic response). The participants were separated into two groups arbitrarily, with a ratio of 60/40 using software. This study involved 521 patients in total. It offered a dependable pre-treatment tool that consists of three blood parameters. The purpose of the work from Neukam et al. (2012) was to identify individuals who're likely to respond to Peg IFN plus RBV treatment, among patients, with HIV/HCV coinfection [24]. As well as, Metwally and AbuSharekh (2018) conducted a study that presented a novel approach utilizing artificial neural networks (ANN) for the purpose of diagnosing the hepatitis virus. Their work described the various factors that have the potential to influence the physical performance of patients. The examination of the gathered data unveiled that the artificial neural network (ANN) model demonstrates a diagnostic accuracy rate of 93% for potential patients [25]. In a recent study, Syafa'ah et al. put out an approach that employs ML classification techniques for the purpose of detecting HCV sickness [26]. With NN algorithms, they were able to attain a 95.12% accuracy rate with success. In addition, Ghazal et al. devised an intelligence system

that utilizes ML algorithms to assist in the diagnosis of hepatitis C [27]. The system demonstrated a high level of accuracy, obtaining an accuracy rate of 97.9% in the staging of HCV. However, utilizing a biochemical test dataset and ML, Ferdib-Al-Islam and Akter classified HCV with an accuracy of 95% employing LR and SVM [28]. Butt et al. proposed a method that uses ML to forecast the human Hepatitis C infection stage by utilizing an ANN algorithm [29]. During evaluation, this diagnostic system achieved a precision rate of 94.44%. Also, Akella and Akella utilized ML algorithms to estimate fibrosis prevalence in patients with hepatitis C achieving an accuracy rate of 81%, in fibrosis estimation [30]. Moreover, the algorithm developed by Barakat et al. employed ML to generate predictions regarding fibrosis in pediatric patients diagnosed with hepatitis C [31]. The researchers utilized the RF algorithm to perform data purification and estimate fibrosis levels subsequent to the completion of the cleansing procedure. Bhingarkar also utilized ML techniques to predict HCV with a 94.3% accuracy rate using the k-NN method [32]. A study from Safdari et al. (2022), in computational health science took an approach to diagnose HCV. They started by preprocessing the data extracted features and finally implemented various ML techniques. Data preprocessing played a role in cleaning and preparing the data for ML algorithms while feature extraction helped simplify the complexity of the data by focusing on essential elements. Once their data was well prepared, Safdari et al. (2022) employed ML techniques. They achieved an accuracy of 95.67% using both LR and k Nearest Neighbors (k-NN). NB yielded a success rate of 92.43% while SVM achieved 94.59%. The DT method scored an accuracy rate of 96.75%. The RF algorithm outperformed them all with an accuracy of 97.29% [33].

## 3 Proposed methodology

This section illustrates the methodology used to evaluate method efficiency in this study. Fig 1 presents an illustration of the methodology utilized. The figure displays a flowchart outlining a method, for classifying HCV infections in patients using ensemble ML. The flowchart illustrates the stages involved including data preprocessing, method training, prediction and evaluation. At the outset, a dataset that contains information about patients with hepatitis C is segmented into subsets with proportions of 80 percent and 20 percent, respectively, for the purposes of training and testing. The training and testing subset undergoes techniques such as feature engineering, handling missing values, undersampling, oversampling as well as feature reduction to create a balanced dataset. This balanced dataset is then utilized for the purpose of training six ML methods that utilize Bagging (bootstrap aggregation) with base classifiers; SVM, RF, LR, k-NN, NB and DT. Once trained, these methods are applied to the testing subset to generate predicted values. These predicted values are then compared against the values using method evaluation metrics, such as accuracy, cross validation (five-fold), precision, recall, Selectivity, F1-score, balanced accuracy, g-mean, standard deviation and error rate. By comparing these results thoroughly, it was determined which method performs best for the classification task. Comparison with previous works in the same field was also conducted.

As outlined in Algorithm 1, the first step in the process involves importing the dataset from a CSV file named HCV-data.csv. This dataset contains relevant attributes for HCV classification. The data is loaded using Python's pandas library to facilitate subsequent preprocessing and analysis. The second step is to conduct an exploratory data analysis (EDA) to gain insights into the dataset's structure and distribution. This includes displaying the dataset's shape, size, and summary statistics, such as mean, median, and standard deviation of numerical features. Additionally, the distribution of the target variable is examined to understand

**Algorithm 1.** Performance comparison of bagging enhanced ML methods on HCV dataset.

1: **Input:** A data set of suspected HCV infections described by $m$ attributes and $n$ instances
2: **Output:** A figure and a bar chart comparing the performance of various bagging enhanced ML methods based on various metrics
3: **Begin**
4: **1. Import Data Set**
5: Load data from "HCV data.csv"
6: **2. Dataset Understanding**
7: Display the shape, size, and summary statistics of the data
8: Display the distribution of the target variable (HCV Category)
9: **3. Feature Engineering**
10: **for** each feature in data **do**
11:    **if** feature is categorical **then**
12:       Convert feature into numerical values using one-hot encoding
13:    **else**
14:       Normalize feature using min-max scaling
15:    **end if**
16: **end for**
17: **4. Handling Missing Values**
18: **for** each feature in data **do**
19:    **if** feature has missing values **then**
20:       Replace missing values with the mode of the feature
21:    **end if**
22: **end for**
23: **5. Feature Selection and Reduction**
24: Apply ANOVA F-statistic test to choose the attributes that are most pertinent to the target variable
25: **6. Get Balanced Dataset**
26: **if** data has class imbalance **then**
27:    Use oversampling and undersampling to balance the data set
28: **end if**
29: **7. Split Data Set**
30: Split data into training_data (80%) and testing_data (20%) using stratified sampling
31: **8. Initialize Methods**
32: methods = [Bagging SVM, Bagging RF, Bagging LR, Bagging KNN, Bagging NB, Bagging DT]
33: **9. Train and Evaluate Methods**
34: **for** each method in methods **do**
35:    Fit method on training data
36:    Predict target variable on testing data
37:    Calculate and display the following metrics
38: **end for**
39: **10. Return** HCV (Active or Inactive)
40: **End**

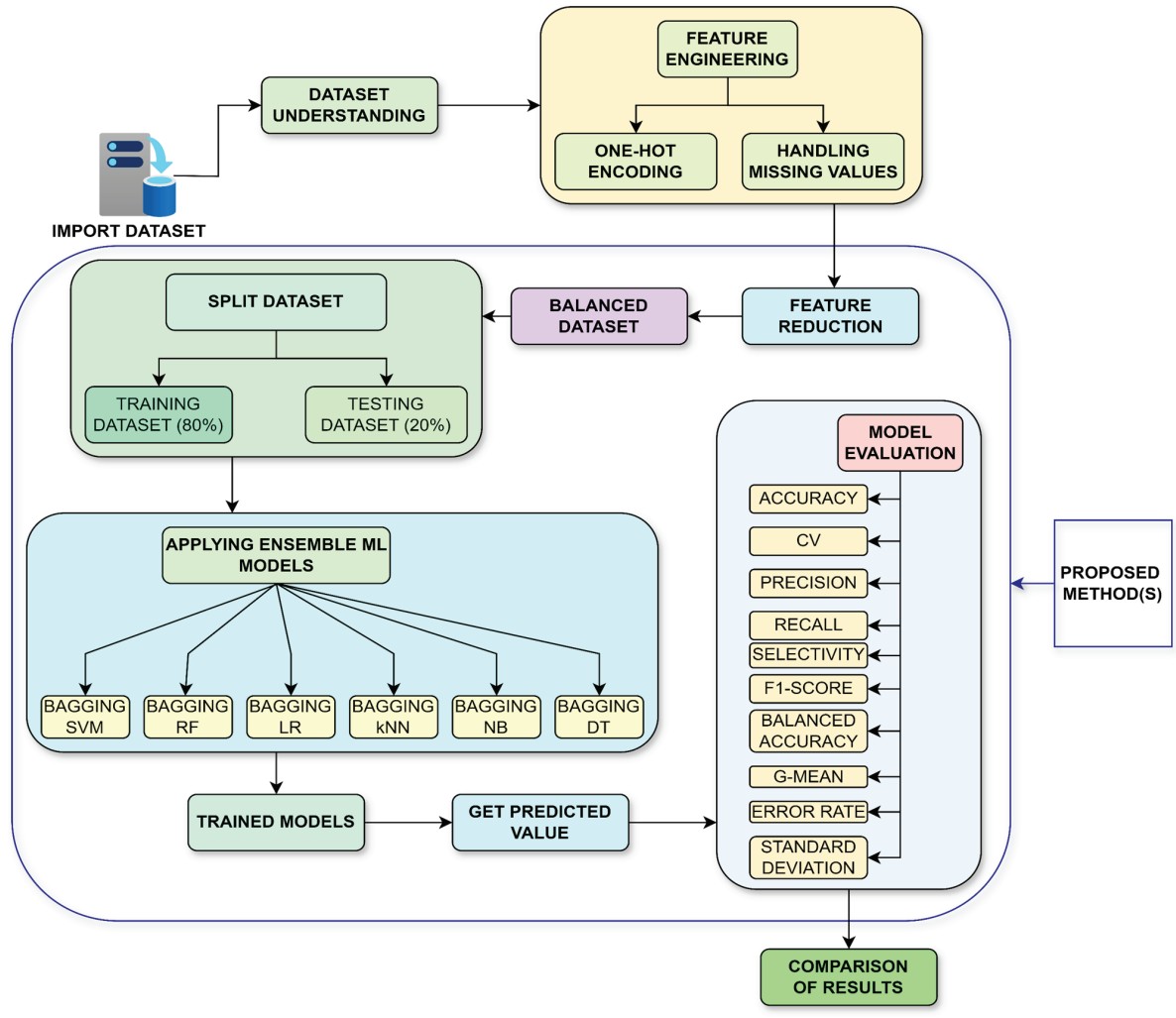

**Fig 1. Proposed methodology.**

the representation of each HCV category, which is essential for designing an effective classification approach. The third step is feature engineering, which involves transforming categorical variables into numerical representations using one-hot encoding. This step is crucial for enabling machine learning models to process categorical data effectively. Moreover, numerical features are normalized using min-max scaling to ensure that all features contribute equally to the model's learning process, preventing any feature from dominating due to differences in scale. The fourth step addresses missing values within the dataset. Missing values are imputed using the mode of the respective feature, ensuring that missing data does not introduce bias or cause errors during model training. This approach is suitable when missing values are relatively few and not randomly distributed. The fifth step involves feature selection and dimensionality reduction to enhance model efficiency and interpretability. The ANOVA F-statistic test is applied to identify the most relevant features with a strong statistical relationship to the target variable. Selecting only the most informative attributes helps improve model performance by reducing noise and computational complexity. The sixth step ensures data balance by checking for class imbalances among HCV categories. An imbalanced dataset can lead to

biased models that favor majority classes while performing poorly on minority classes. If an imbalance is detected, oversampling and undersampling techniques are employed. Oversampling increases the instances of minority classes, while undersampling reduces instances of majority classes, leading to a more balanced and fair representation of all categories. The seventh step is to split the dataset into training and testing subsets using stratified sampling. A stratified approach ensures that each subset maintains the same proportion of HCV categories as in the original dataset, preventing bias in model training. An 80–20 split is chosen for training and testing, meaning that 80% of the data is allocated for model training, while 20% is reserved for performance evaluation. This ratio is selected based on a trade-off between model generalization and training stability. A higher training proportion (e.g., 90–10) lead to overfitting, where the model learns patterns specific to the training data but fails to generalize well to unseen instances. Conversely, a lower training proportion (e.g., 70–30 or 60–40) reduces the amount of data available for learning, potentially leading to underfitting and poor performance. The 80–20 ratio strikes a balance, providing sufficient data for training while ensuring a reliable evaluation set. The eighth step initializes machine learning models by implementing a list of bagging-based ensemble methods. These include Bagging SVM, Bagging RF, Bagging LR, Bagging k-NN, Bagging NB, and DT. Bagging is used to enhance model robustness by training multiple weak learners on different bootstrap samples and aggregating their predictions to reduce variance and improve accuracy. The ninth step involves training and evaluating each machine learning method. This is achieved by iterating through the initialized models and performing the following actions: fitting the model on the training data, predicting target labels on the test data, and computing various performance metrics. The metrics used for evaluation include accuracy, cross-validation score (CV), precision, recall, specificity, balanced accuracy, geometric mean (G-mean), error rate, and F1-score. These metrics provide a comprehensive assessment of each model's effectiveness in handling the classification task. The tenth and final step compares the results of different machine learning models. A comparative analysis is performed using a table and bar chart to visualize and compare the performance of each model across multiple metrics. This allows for a clear and intuitive assessment of which models perform best in terms of accuracy, robustness, and class balance. The final table and bar chart serve as a reference for selecting the most suitable model for HCV classification.

## Implementations of the classifications methods

This subsection elaborates on the utilization of classification ML methods used in this study. The ML methods used here are SVM, k-NN, LR, RF, NB, and DT which are formally acknowledged as supervised ML methods. They were operationalized to categorize pre-existing data in a medical dataset. The collected dataset was judiciously divided into a training segment (comprising 80% of the total data) and a testing segment (accounting for the residual 20%). Following this partitioning, each classification model was subjected to a learning process utilizing balanced training data. Upon the completion of the learning phase, the classifier proceeded to categorize patients, deriving this classification from the records contained within the testing dataset. The previously described test data was then used to assess the classifiers' effectiveness. Using several measures, the performance quotient of every model was calculated. Subsequently, leveraging Python version 3, the evolution of these models was facilitated by diverse classifiers. The specifics of these have been described in this study report in the subsequent sections, in addition to this, an integral component of the study involved the utilization of Bagging, a meta-algorithm, in combination with six distinctive algorithms. These comprise the SVM, Gaussian NB, DT, RF, LR, and the k-NN algorithm. Each algorithm

imparts a unique facet to the overall model, providing an exhaustive and meticulous approach towards data classification.

**Bagging (Bootstap Aggregation).** Bagging, an abbreviation for Bootstrap Aggregating, constitutes a potent technique, instrumental in enhancing the precision and reliability of a predictive model, while retaining applicability to a broad spectrum of model types. Bagging operates on the principle of generating multiple derived subsets from the principal dataset via a method of random sampling with replacement. Each of these discrete subsets is employed to educate an individual model, with the ultimate prediction being an aggregate of the predictions rendered by every model. Bagging promotes model robustness by reducing the variance and potential overfitting, thereby enhancing the overall performance of the composite model.

Let $D$ be the original training set of size $n$, and let $Q(X, Y \mid D)$ be a probability distribution that picks a training sample $(x_i, y_i)$ from $D$ uniformly at random. Then, sample $m$ subsets of data $D_1, D_2, \ldots, D_m$ can be sampled from $Q^n$, where each subset has the same size $n$ as the original set and is sampled with replacement.

For each subset $D_j$, a model $h_j$ is trained on it using some base learning algorithm, such as a decision tree (DT). The model $h_j$ can be a classifier or a regressor, depending on the task.

- The final prediction for a new input $x$ is obtained by averaging the predictions of all the models:

$$h(x) = \frac{1}{m} \sum_{j=1}^{m} h_j(x). \tag{1}$$

- If the models are classifiers, majority voting can be used instead of averaging:

$$h(x) = \mathrm{mode}(h_1(x), h_2(x), \ldots, h_m(x)). \tag{2}$$

Though frequently utilized in the context of DTs, Bagging maintains its versatility and can be extended to a myriad of model types. This includes, but is not limited to, logistic regression (LR), random forests (RF), and support vector machines (SVMs). Through the amalgamation of various model predictions, Bagging has the potential to curtail model variance and augment its overall capacity for accurate generalization.

**Bagging decision tree.** In Bagging with DT, a random sampling with replacement technique is used to train several DTs on distinct portions of the training data. The final result is generated by averaging the predictions of all the trees, each of which is trained on a distinct subset of the data. This enhances the model's accuracy and lessens overfitting.

Bagging with DT has been used in various applications such as credit scoring, breast cancer classification, network intrusion detection [39], and diabetes risk prediction. In these applications, bagging with DT has been shown to improve the accuracy of the models and reduce overfitting.

**Explanation of each step in the Fig 2 flowchart:**

1. **Start:** This is the starting point of the Bagging process.
2. **Split the training data into N subsets using random sampling with replacement:** In this stage, random sampling with replacement is used to split the training data into $N$ subsets, or samples. Since it's a replacement dataset, certain data points may be repeated in each subset while others may be omitted. Each subset is formed by randomly choosing data points from the original dataset. There is variability among the subsets thanks to this random picking.

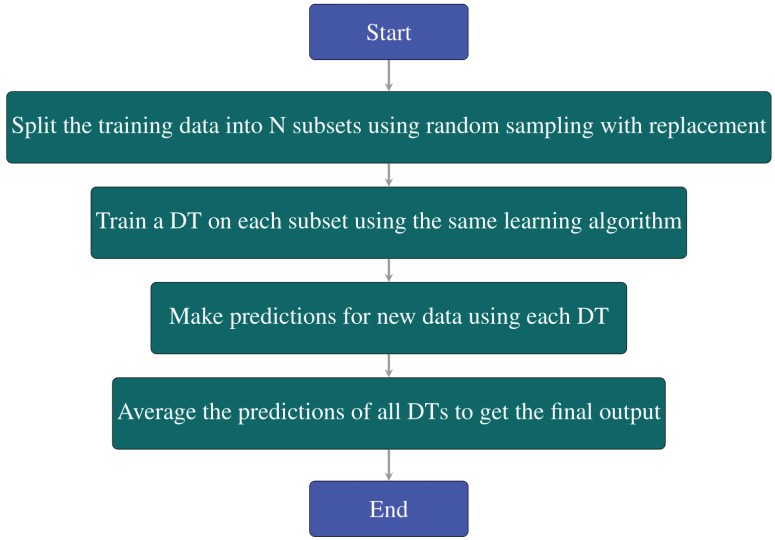

**Fig 2. Flowchart of bagging DT.**

3. **Train a Decision Tree (DT) on each subset using the same learning algorithm:** For each of the *N* subsets created in the previous step, a Decision Tree (DT) is trained using the same learning algorithm. This means that each subset is used to train an individual Decision Tree, but all Decision Trees are built using the same algorithm.

4. **Make predictions for new data using each DT:** Once the *N* Decision Trees have been trained, they are used to make predictions for new or unseen data. Each Decision Tree generates a unique prediction based on the input data provided.

5. **Average the predictions of all DTs to get the final output:** In this step, the final output or prediction is produced by averaging or combining the predictions from each individual Decision Tree. This averaging reduces variance and improves the overall accuracy of the prediction. For classification tasks, this could involve taking a majority vote, while for regression tasks, it involves averaging the numerical predictions.

6. **End:** This is the end of the Bagging process.

**Bagging support vector machine.** Bagging with SVM is a widely used and effective technique to improve the accuracy and stability of SVM models, especially for high-variance and complex problems such as classification or regression. It is an ensemble learning technique that integrates the predictions of multiple SVM models trained on different subsets of the same dataset using bootstrap sampling.

**Explanation of each step in the Fig 3 flowchart:**

1. **Initialize Ensemble:** Initialize an ensemble to store multiple Support Vector Machine (SVM) models.

2. **For *i* = 1 to *N* (Number of SVMs):** This loop runs *N* times, where *N* is the number of SVM models to be trained.

3. **Sample Dataset with Bootstrap Sampling:** In each iteration, the training dataset is sampled with replacement using bootstrap sampling. This creates a new subset of the training data for training each SVM model.

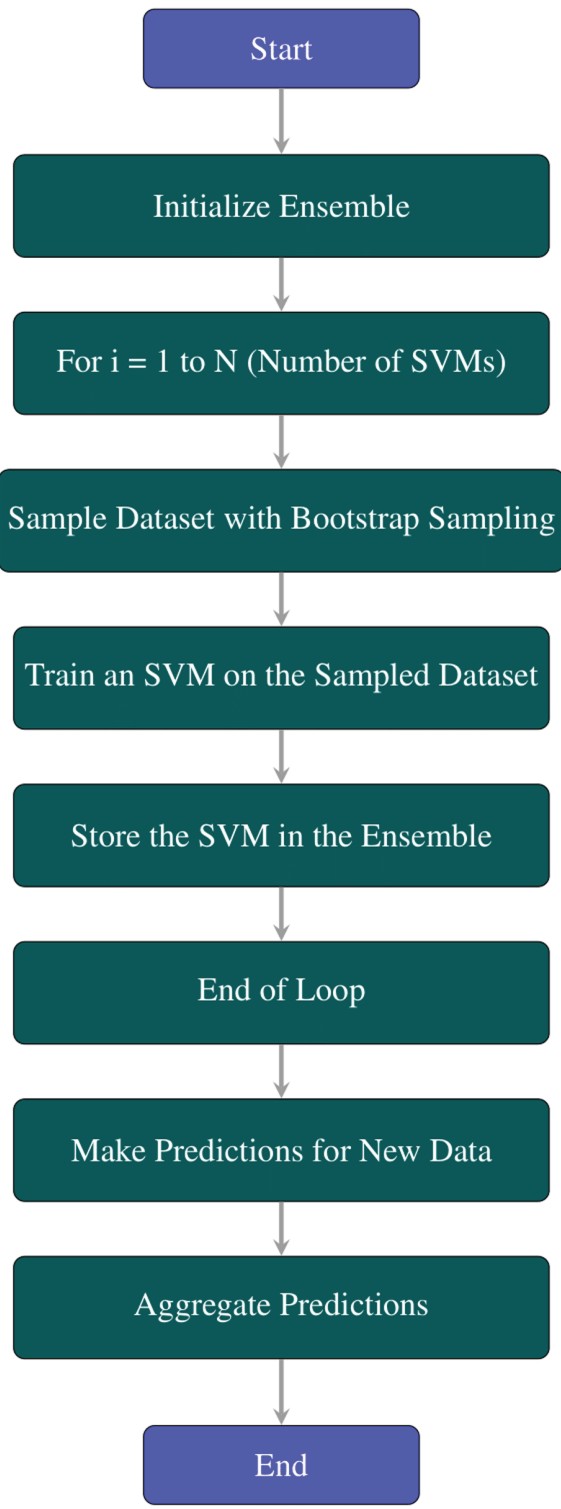

**Fig 3. Flowchart of bagging SVM.**

4. **Train an SVM on the Sampled Dataset:** Train an SVM model on the sampled dataset from step 3. Each SVM is trained on a different subset of the data.

5. **Store the SVM in the Ensemble:** Add the trained SVM model to the ensemble of models.

6. **End of Loop:** Repeat steps 3–5 until $N$ SVM models have been trained.

7. **Make Predictions for New Data:** Use the ensemble of SVM models to make predictions for new or unseen data points.

8. **Aggregate Predictions:** Aggregate the predictions made by each SVM model. Techniques like voting (for classification tasks) and averaging (for regression tasks) can be used to combine the predictions.

9. **End:** End of the Bagging process with SVM.

**Bagging random forest.** The application of Bagging in the context of RF, a celebrated methodology that draws upon both Bagging and DTs. Under this scheme, the foundational methods are DTs, each trained on distinctive feature and instance subsets obtained via bootstrap sampling. The integration of Bagging with RF constitutes a frequently used and efficacious approach for amplifying the precision and robustness of DT methods. This technique is particularly valuable when addressing issues characterized by high variance and complexity, such as classification or regression tasks. The standard procedure involves training DTs on distinct subsets of features and instances, a process facilitated through bootstrap sampling. The collective decision is then obtained either through majority voting or averaging the predictions. This strategy is capable of tackling a wide range of problems, including but not limited to, classification, regression, and clustering.

**Explanation of each step in the Fig 4 flowchart:**

1. **Initialize Random Forest:** Initialize a Random Forest ensemble to store multiple Decision Tree (DT) models.

2. **For $i = 1$ to $N$ (Number of DTs):** This loop runs $N$ times, where $N$ is the number of DT models to be created.

3. **Sample Features with Bootstrap Sampling:** In each iteration, sample a subset of features from the dataset with replacement using bootstrap sampling. This ensures that different features are used to train each DT model.

4. **Sample Instances with Bootstrap Sampling:** In each iteration, also sample a subset of instances (data points) from the dataset with replacement using bootstrap sampling. This creates a unique training dataset for each DT.

5. **Train a DT on the Sampled Dataset:** Train a DT model on the sampled dataset, which consists of the selected features and instances.

6. **Store the DT in the Random Forest:** Add the trained DT model to the Random Forest ensemble.

7. **End of Loop:** Repeat steps 3–6 until $N$ DT models have been trained.

8. **Make Predictions for New Data:** Use the Random Forest ensemble to make predictions for new or unseen data points. Each DT in the ensemble will provide its own prediction.

9. **Aggregate Predictions:** Aggregate the predictions made by each DT model. Methods such as averaging (for regression tasks) or majority voting (for classification tasks) can be used to combine the predictions.

10. **End:** End of the Bagging process with Random Forest.

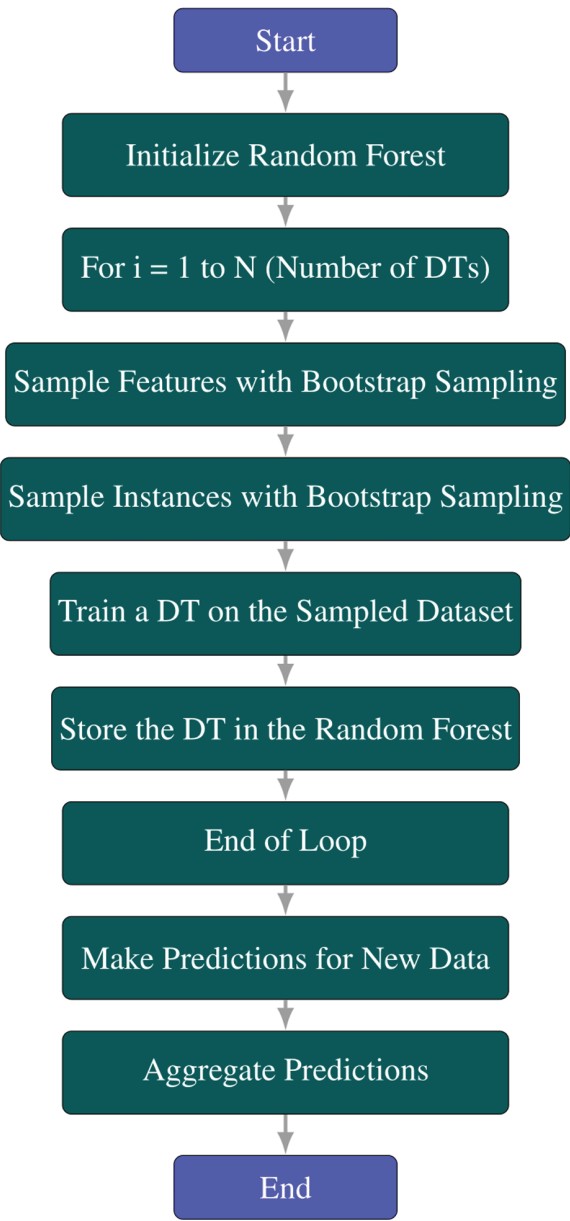

**Fig 4. Flowchart of bagging RF.**

**Bagging logistic regression.** LR is a straightforward and speedy probabilistic classifier that rests on the assumption of feature independence and their linear relation to the log-odds of a binary outcome variable. The methodology of Bagging in LR involves training distinct LRs on varying instance subsets, a process facilitated through bootstrap sampling. The collective decision is then determined through majority voting or the averaging of predictions.

**Explanation of each step in the Fig 5 flowchart:**

1. **Initialize Logistic Regression:** Initialize a Logistic Regression (LR) method.
2. **For** $i = 1$ **to** $N$ **(Number of LRs):** This loop runs $N$ times, where $N$ is the number of LR models to be created.

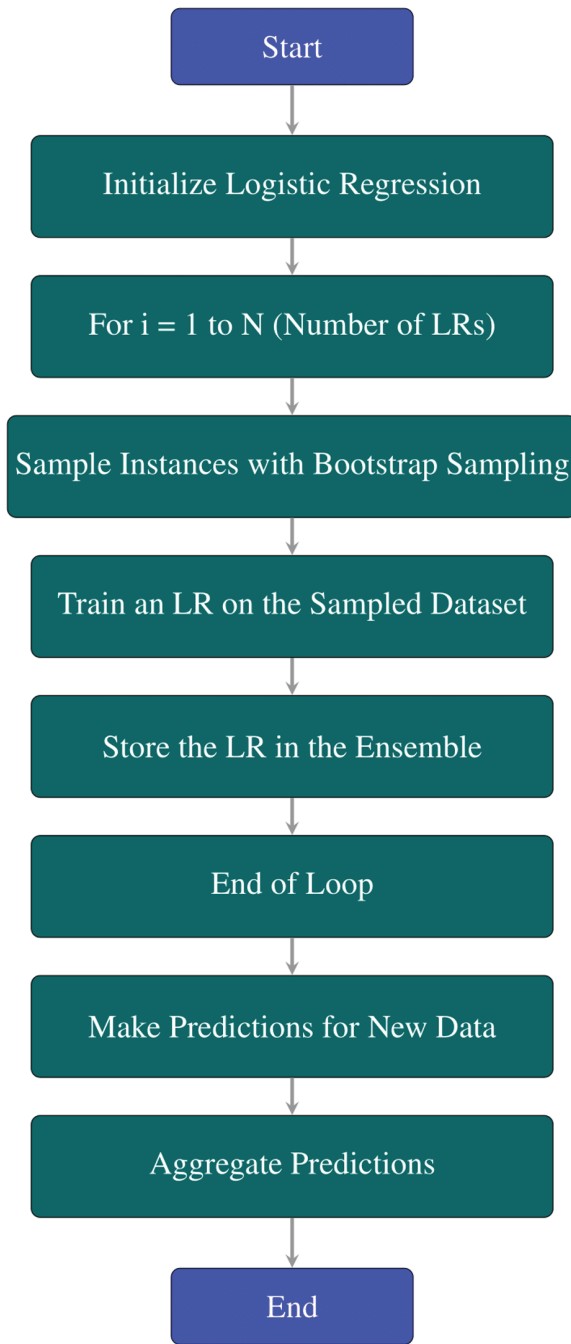

**Fig 5. Flowchart of bagging LR.**

3. **Sample Instances with Bootstrap Sampling:** In each iteration, sample a subset of instances (data points) from the dataset with replacement using bootstrap sampling. This creates a new dataset for training each LR model.

4. **Train an LR on the Sampled Dataset:** Train a Logistic Regression model on the sampled dataset.

5. **Store the LR in the Ensemble:** Add the trained LR model to the ensemble of Logistic Regression methods.
6. **End of Loop:** Repeat steps 3–5 until $N$ Logistic Regression models have been trained.
7. **Make Predictions for New Data:** Use the ensemble of LR models to make predictions for new or unseen data points.
8. **Aggregate Predictions:** Aggregate the predictions made by each LR model. Techniques such as averaging (for regression tasks) or majority voting (for classification tasks) can be used to combine the predictions.
9. **End:** End of the Bagging process with Logistic Regression.

**Bagging Gaussian Naïve Bias.** Gaussian NB is a classifier that is both straightforward and swift in its operation. This probabilistic classifier operates under the assumption of feature independence, with each feature following a normal distribution when considered in light of the class label. Employing Bagging with Gaussian NB is an uncomplicated process, wherein each Gaussian NB method is trained on a unique subset of instances derived through bootstrap sampling. The final predictions are then synthesized through majority voting or averaging mechanisms.

**Explanation of each step in the Fig 6 flowchart:**

1. **Initialize Gaussian NB Classifier:** Initialize a Gaussian Naive Bayes (NB) classifier.
2. **For $i = 1$ to $N$ (Number of NBs):** This loop runs $N$ times, where $N$ is the number of Gaussian NB classifiers to be created.
3. **Sample Instances with Bootstrap Sampling:** In each iteration, sample a subset of instances (data points) from the dataset with replacement using bootstrap sampling. This creates a unique dataset for training each Gaussian NB model.
4. **Train a NB on the Sampled Dataset:** Train a Gaussian NB model on the sampled dataset from step 3.
5. **Store the NB in the Ensemble:** Add the trained Gaussian NB model to the ensemble of classifiers.
6. **End of Loop:** Repeat steps 3–5 until $N$ Gaussian NB classifiers have been trained.
7. **Make Predictions for New Data:** Use the ensemble of Gaussian NB models to make predictions for new or unseen data points.
8. **Aggregate Predictions:** Aggregate the predictions made by each Gaussian NB model. Techniques such as majority voting (for classification tasks) or averaging (for regression tasks) can be used for this purpose.
9. **End:** End of the Bagging process with Gaussian NB classifiers.

## Bagging k-nearest neighbors

The k-NN algorithm is an intuitively appealing non-parametric classifier. It works on a principle of proximity, associating a new instance with the class most frequent among its 'k' nearest neighbors in the feature space [43]. Enriching the k-NN approach with Bagging involves creating multiple k-NN methods, each trained on a distinct subset of instances procured through bootstrap sampling. The collective predictions from these methods are consolidated through a process of majority voting or averaging, leading to the final decision.

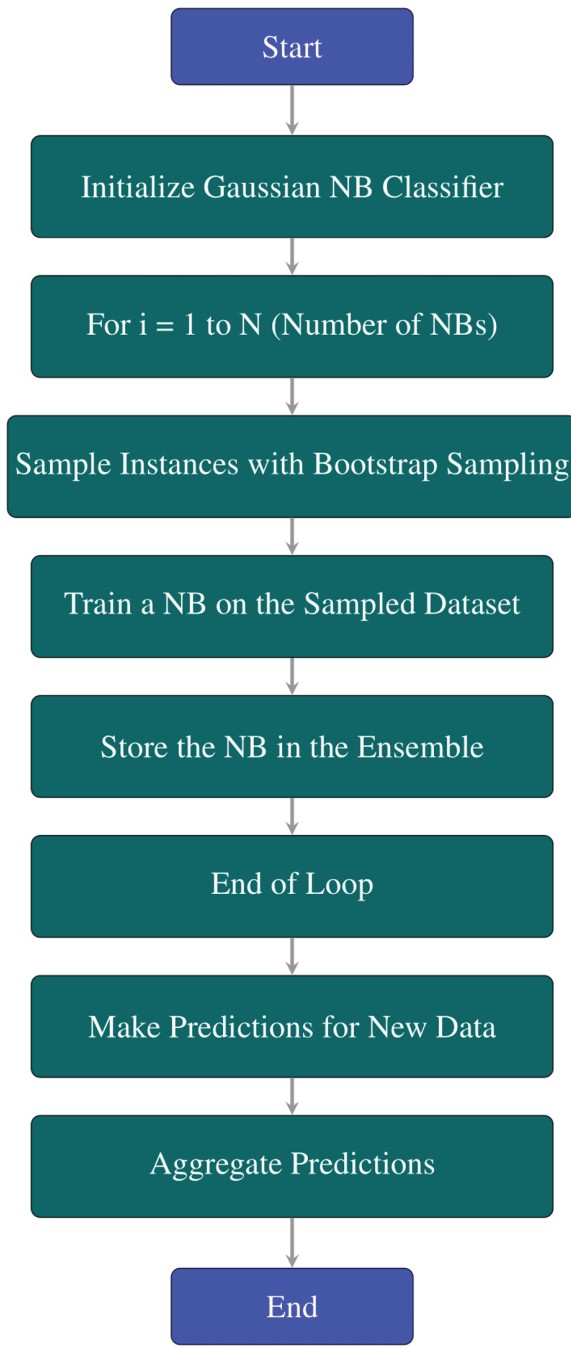

**Fig 6. Flowchart of bagging NB.**

**Explanation of each step in the Fig 7 flowchart:**

1. **Initialize k-NN Classifier:** Initialize a k-Nearest Neighbors (k-NN) classifier.
2. **For** $i$ = 1 **to** $N$ **(Number of k-NNs):** This loop runs $N$ times, where $N$ is the number of k-NN classifiers to be created.

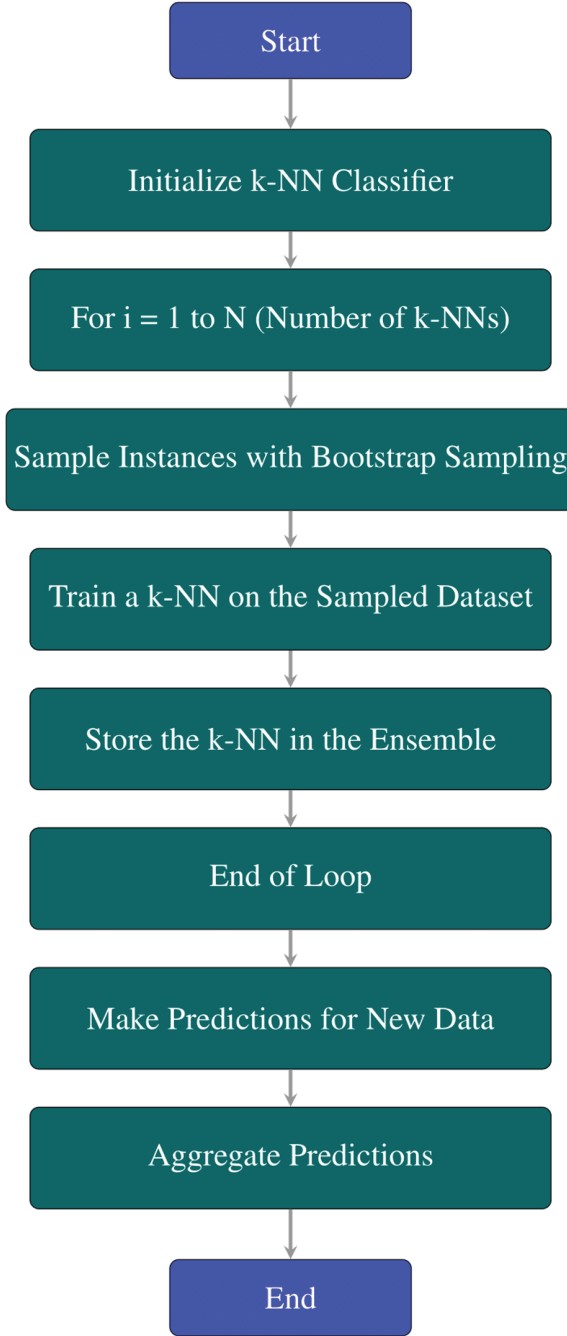

**Fig 7. Flowchart of bagging k-NN.**

3. **Sample Instances with Bootstrap Sampling:** In each iteration, sample a subset of instances (data points) from the dataset with replacement using bootstrap sampling. This creates a new dataset for training each k-NN model.

4. **Train a k-NN on the Sampled Dataset:** Train a k-NN model on the sampled dataset from step 3.

5. **Store the k-NN in the Ensemble:** Add the trained k-NN model to the ensemble of classifiers.

6. **End of Loop:** Repeat steps 3–5 until $N$ k-NN classifiers have been trained.

7. **Make Predictions for New Data:** Use the ensemble of k-NN models to make predictions for new or unseen data points.

8. **Aggregate Predictions:** Aggregate the predictions made by each k-NN model. Methods such as averaging (for regression tasks) or majority voting (for classification tasks) can be used to combine the predictions.

9. **End:** End of the Bagging process with k-NN classifiers.

## Dataset description

The University of California, Irvine (UCI) ML Repository provided access to the HCV dataset, which was utilized in this study [34]. The said dataset encapsulates both clinical laboratory results and demographic details for a sample size of 615, inclusive of blood donors and hepatitis C patients.

Each individual record within this dataset is characterized by 12 specified variables. These variables pertain to a range of medical tests and demographic indicators. The medical tests encompass the measurement of aspartate aminotransferase (AST), alanine aminotransferase (ALT), albumin (ALB), bilirubin (BIL), cholesterol (CHOL), creatinine blood test (CREA), choline esterase (CHE), $\gamma$-glutamyl-transferase (GGT) , and total protein test (PROT). As for demographic traits, the dataset also includes variables for age and sex. Blood donors, suspected blood donors, people with HCV, fibrosis, and cirrhosis are the five possible diagnostic outcomes included in the dataset's final analysis and diagnosis. The spectrum of HCV diagnoses among patients ranges from asymptomatic chronic HCV infection to end-stage liver cirrhosis necessitating liver transplantation (LTx). Following the preprocessing phase, the dataset comprised 615 instances. The details of the 12 variables are outlined in Table 1. For every entry in the dataset, there were ten standard diagnostic test results for liver disorders. The patients ranged in age from 19 to 77 years, with a mean age of 47.40 ± 10.05 years.

**Table 1. Dataset variables described (in Raw Form).**

| Variables | Interpretation | Data class |
|---|---|---|
| 1. Age | The patients' age in years | Integer |
| 2. Sex | Gender distribution among patients | Categorical |
| 3. ALB | Blood albumin content | Real |
| 4. ALP | Alkaline phosphatase enzyme level in blood | Real |
| 5. ALT | Alanine aminotransferase level indicating liver damage | Real |
| 6. AST | Aspartate aminotransferase level found in liver and muscles | Real |
| 7. BIL | Bilirubin level in the blood | Real |
| 8. CHE | Serum cholinesterase level reflecting liver function | Real |
| 9. CHOL | Cholesterol and triglycerides level in the blood | Real |
| 10. CREA | Creatinine level measuring kidney function | Real |
| 11. GGT | Gamma-glutamyl transferase level related to liver disease | Real |
| 12. PROT | Total protein level in the blood | Real |

The classification outcome was the disease category. Patients with hepatitis (3.90%), fibrosis (3.41%), and cirrhosis (3.41%) made up the majority of the study participants, followed by those who donated blood (86.67%) and probable blood donors (1.14%). The goal of the dataset was to distinguish between healthy people and hepatitis patients, hence the outcome variable was categorized as a true/false value. This binary variable might either be non-hepatitis (including blood donors and suspected blood donors) or hepatic (at three different stages) for maximum performance. After feature reduction, correlations were computed between the outcome variable and all clinical factors in the development of supervised learning algorithms. The correlation coefficient is a useful statistic for studying the association between independent variables and the categories they describe.

**Correlation of dataset variables.** A heat map is a graphical representation of data where the values of a matrix are encoded by colors. A heat map can reveal patterns or correlations in the data by visualizing the relative differences between the values.

The heat map, displayed in Fig 8, displays the association between several variables in this dataset. There was a 1:4 split between the training and testing portions of the dataset. The dataset was unbalanced, as reflected from the arrangement of patients' records. To address this imbalance, a combination of Oversampling and Undersampling techniques was employed instead of the widely used SMOTE algorithm. After the application of these techniques, the newly processed data was ready for use. To prevent data leakage and reduce the risk of method overfitting, these techniques were applied solely to the training set. Furthermore, to enhance the method's performance given the class imbalance, ensemble ML methods were utilized, specifically bagging. This approach helps to improve the preciseness of ML algorithms by reducing variance and helping to avoid overfitting.

In this case, Fig 8 shows the correlation matrix of the independent variables in the HCV dataset. These variables include measures such as albumin (ALB), alkaline phosphatase (ALP), alanine aminotransferase (ALT), aspartate aminotransferase (AST), cholinesterase (CHE), creatinine (CREA), gamma-glutamyl transferase (GGT), and total protein (PROT). The correlation coefficients range from –1 to 1, indicating the strength and direction of the linear relationship between pairs of variables.

**Analysis of Correlations:**

1. **Strongest Positive Correlation:**
   - **ALB and PROT:** The strongest positive correlation is between ALB (Albumin) and PROT (Total Protein), with a correlation coefficient of 0.55. This suggests a moderate to strong linear relationship where higher levels of albumin are associated with higher total protein levels in the blood.
2. **Strongest Negative Correlation:**
   - **AST and CHE:** The strongest negative correlation is between AST (Aspartate Aminotransferase) and CHE (Cholinesterase), with a correlation coefficient of –0.21. This implies that as the level of AST increases, the level of CHE tends to decrease, though the relationship is relatively weak.
3. **Other Notable Correlations:**
   - **GGT and AST:** GGT (Gamma-Glutamyl Transferase) and AST have a positive correlation of 0.49, indicating a moderate positive relationship.
   - **ALB and CHE:** ALB and CHE have a positive correlation of 0.38, suggesting that as albumin levels increase, cholinesterase levels also tend to increase moderately.
   - **ALP and GGT:** There is a moderate positive correlation of 0.42 between ALP (Alkaline Phosphatase) and GGT, indicating that higher levels of ALP are associated with higher levels of GGT.

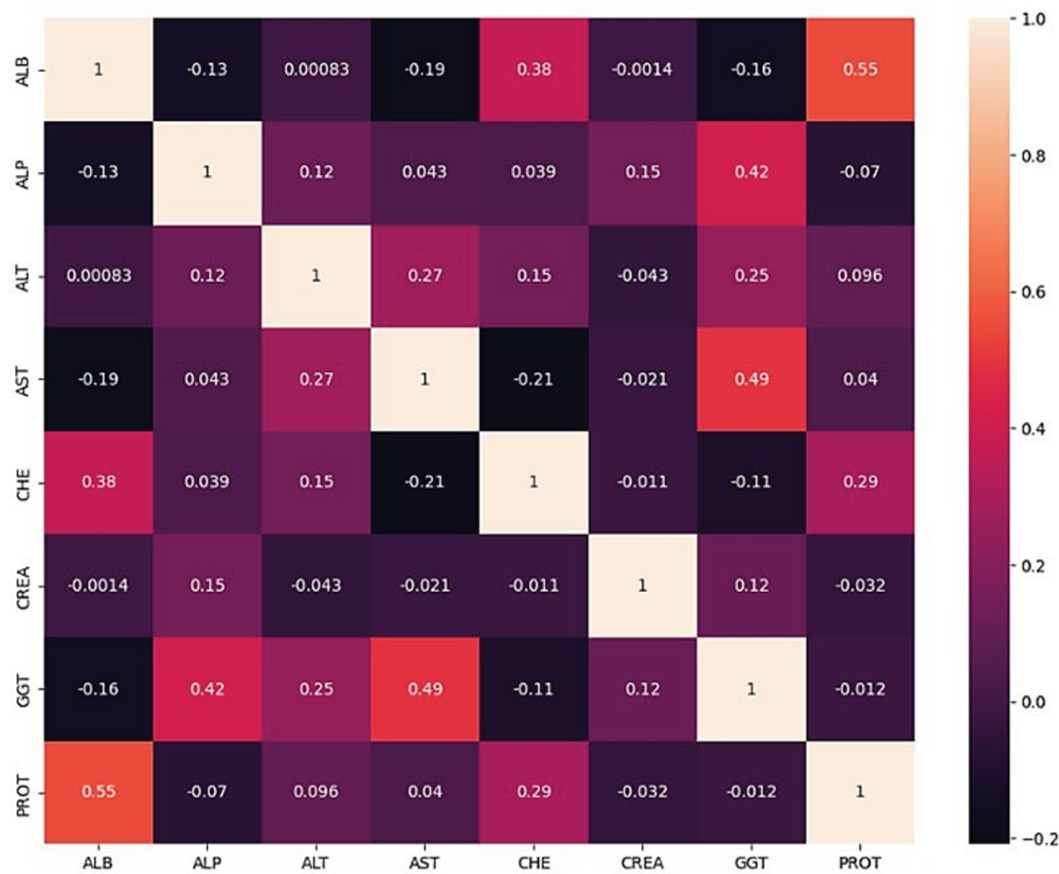

**Fig 8. Correlation of dataset variables.**                                                                    AQ1

4. **Weak and Negligible Correlations:**
   - Many pairs show very weak correlations, close to zero, indicating little to no linear relationship. For instance, the correlation between CREA and most other variables (like ALT, AST, and GGT) is close to zero, with coefficients ranging from –0.012 to 0.15.

## Feature engineering

In the process of refining the methodology, this study incorporated a feature selection and reduction technique by utilizing one-way ANOVA-F statistic testing. This strategy of feature reduction is an integral aspect of data preprocessing, which intends to determine a subset of suitable features for subsequent application in the construction of methods. Feature reduction has multiple goals, including the simplification of methods, enhancement of data mining performance, and the provision of clean and interpretable data. In the realm of big data mining, the significance of feature selection is particularly emphasized due to the formidable challenges and opportunities associated with high-dimensional and intricate data. Among the various feature engineering techniques applicable for numerical input data and categorical target variables, one-way ANOVA-F statistic has gained substantial prominence. ANOVA,

or analysis of variance, is a statistical test that ascertains the differences in the means of distinct groups of a variable. The F-statistic gauges the ratio of between-group variance to the within-group variance.

The F-statistic indicates a greater disparity between group means, signifying that the corresponding feature has a stronger association with the target variable. The ANOVA F-statistic is particularly effective in ranking features based on their significance, facilitating the selection of the most relevant attributes for data analysis and classification.In this study, the ANOVA F-test was applied to assess the statistical relevance of each feature concerning the identification of HCV-infected patients. Features that exhibited minimal variation across different HCV categories, and therefore had low F-statistics, were deemed irrelevant to the classification task. As a result, attributes such as age, sex, bilirubin, and cholesterol were removed, as their inclusion did not contribute meaningfully to distinguishing between HCV-infected and non-infected individuals.The elimination of these attributes aligns with existing clinical research. For instance, Fernández Salazar et al. reported that hepatic steatosis, a liver condition frequently observed in chronic hepatitis C patients, does not correlate with age, sex, or cholesterol levels [35].Additionally, there is no established clinical linking HCV infection to demographic variables such as age, sex, or physiological markers like BMI, blood pressure, bilirubin, and cholesterol. Given that these attributes have not been recognized as significant diagnostic indicators of HCV infection in medical literature, their exclusion ensures that the model focuses on features with stronger predictive power. Furthermore, in the current dataset, the results of the ANOVA F-test reaffirmed this finding by demonstrating that age, sex, bilirubin, and cholesterol had negligible impact on the classification of HCV infection. Their low F-statistics suggested that these attributes did not provide discriminative information, making their removal beneficial for improving model efficiency and reducing potential noise in the data. Consequently, only the most clinically relevant and statistically significant features were retained, ensuring that the predictive model is both data-driven and aligned with established medical knowledge.

**Dataset features' significance based on P-values acquired from ANOVA-F test.** The Fig 9 illustrates the significance of various features based on their p-values, with a significance threshold set at 0.05. Features with p-values above this threshold are considered not significant and are highlighted in red, indicating their removal from the dataset. These removed features include Age (p-value: 0.486), Sex (p-value: 0.181), Bilirubin level (BIL) (p-value: 0.059), and Cholesterol and triglycerides level (CHOL) (p-value: 0.364). On the other hand, features with p-values below the threshold, deemed significant and shown in green, are retained for further analysis. These retained features are Blood albumin content (ALB)
(p-value: 0.000), Alkaline phosphatase enzyme level (ALP) (p-value: 0.043), Alanine aminotransferase level (ALT) (p-value: 0.046), Aspartate aminotransferase level (AST) (p-value: 0.000), Serum cholinesterase level (CHE) (p-value: 0.000), Creatinine level (CREA) (p-value: 0.037), Gamma-glutamyl transferase level (GGT) (p-value: 0.000), and Total protein level (PROT) (p-value: 0.013). The bar chart effectively differentiates between significant and non-significant features, aiding in the selection of relevant variables for subsequent analysis. However, the exclusion of demographic factors such as age and sex raises an important consideration regarding potential biases within the dataset. While these attributes were found to be statistically insignificant in distinguishing between HCV-infected and non-infected individuals, their lack of significance could stem from an imbalanced dataset, where the distribution of age and gender is not representative of a broader population.

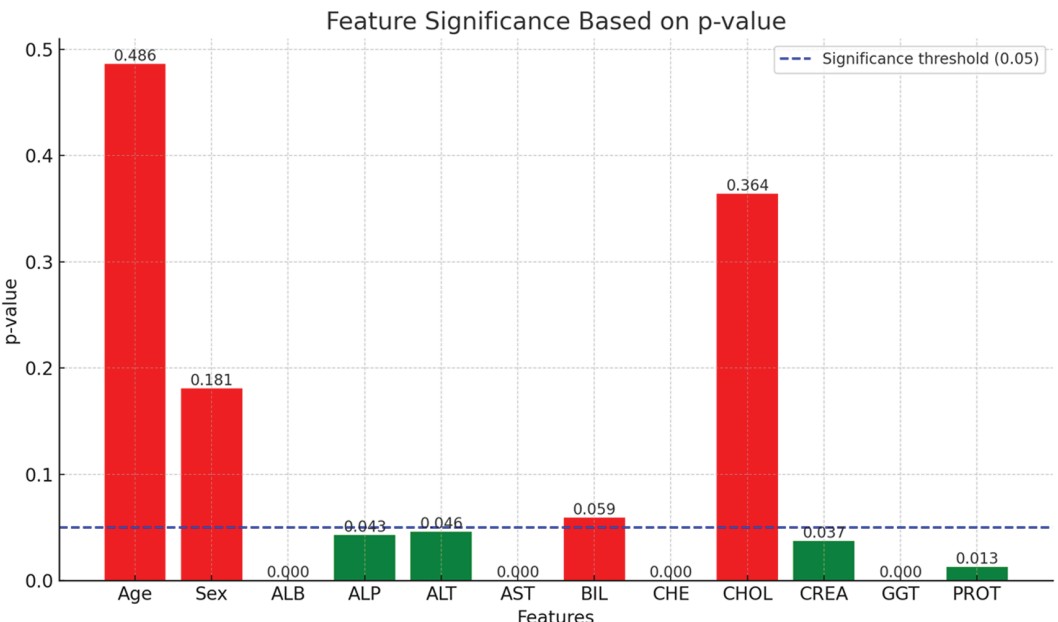

**Fig 9. P- P-value for different features.**

## Data refinement process

The data excavated from patient records frequently lack comprehensive clarity, necessitating a thorough cleansing process as a critical precursor to ML method development. The preprocessing of data consists of translating raw data into a coherent and interpretable format. This ensures a uniform value range across the dataset and establishes comparability amongst features. Therefore, the raw data underwent normalization, restructuring it into appropriate forms amenable to disparate ML estimators. Initially, the ID column was expunged. Any instances of missing values within the dataset were supplanted by the MODE value of each corresponding variable. To address the issue of varying measuring units, the process of data normalization was performed using the StandardScaler function. This function is designed to scale variables to a unit variance.

In data mining and ML, class imbalance occurs when there is an unequal distribution of data points between different classes. This inequity typically arises when one class contains a disproportionately large number of instances compared to the other(s). The term "majority class" is used to describe the more common category, whereas "minority class" describes the rarer category. In the context of data refinement or preprocessing, class imbalance can pose significant challenges. The goal of many ML algorithms is to achieve the highest possible level of accuracy, which can result in a favoritism for the more common class. As a consequence of this, the performance of these algorithms is frequently subpar when applied to the minority class. In the data mining field in research, class imbalance is addressed during the data refinement process, which involves preprocessing the data to improve the quality and reliability of the subsequent analysis. In the context of this study, an imbalance was present in the dataset, indicating that a majority of records were aligned to a single category, as depicted in Fig 10. Classification tasks with such skewed datasets typically exhibit a bias towards larger categories.

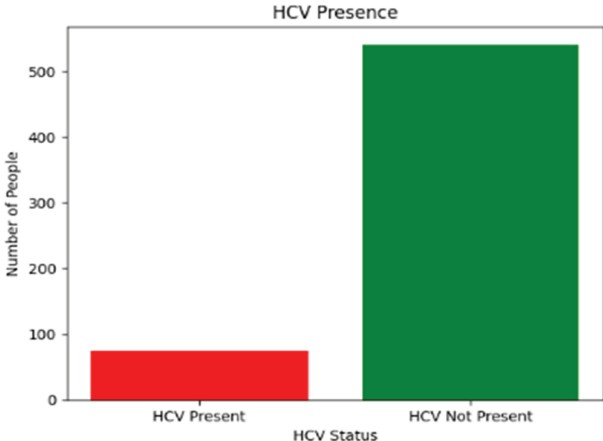

**Fig 10. Distribution of HCV presence in dataset before data refinement (Oversampling and Undersampling).**

To counteract this, both oversampling and undersampling techniques were applied to the HCV dataset, potentially enhancing the performance fluency of various classifiers ML algorithms. In essence, oversampling helps to remedy the imbalance by increasing the representation of the underrepresented minority class, as illustrated in Fig 12. This is accomplished by either producing artificial cases for the data or simply replicating existing examples. On the contrary, undersampling reduces the samples from the majority class to achieve a balance between classes. As depicted in Fig 11, this procedure ensures that the smaller category receives adequate representation during the classification process, while also preventing the method from being overwhelmed by the larger category. By using both oversampling and undersampling techniques, this study achieves a balanced dataset that yields more accurate and unbiased results.

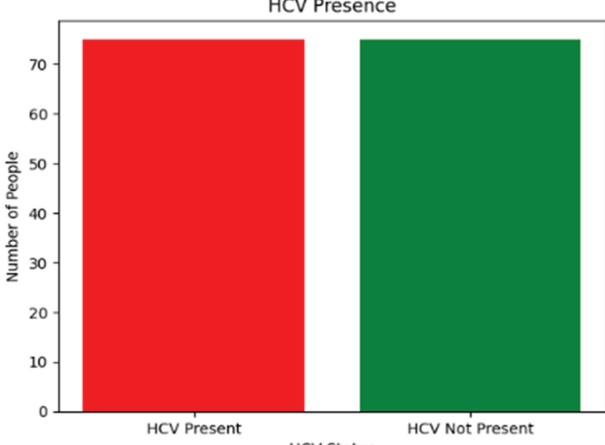

**Fig 11. Distribution of HCV presence in dataset after undersampling.**

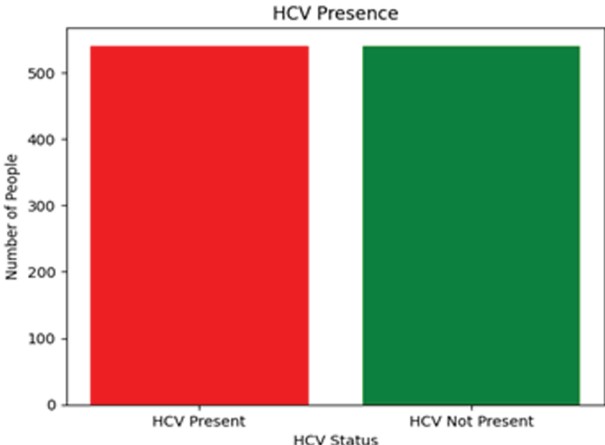

**Fig 12. Distribution of HCV presence in dataset after oversampling.**

**Oversampling.**  Oversampling is a resampling technique that alters the distribution pattern of a variable within the dataset by synthetically augmenting the total number of observations that possess a specific value or a range of values for that variable. Oversampling is frequently employed to address imbalanced classification problems, where a particular class is considerably underrepresented in the data compared to another class. As depicted in Fig 13, the steps of oversampling are as follows:

1. Specify the variable and the value or the range of values that require oversampling. This is typically the target variable and the minority class label.
2. Establish the preferred ratio or proportion of the oversampled value or range of values to the other values. This can depend on factors such as equalizing the class distribution or attaining a certain performance measure.
3. Randomly choose observations from the original dataset that have the oversampled value or range of values, with or without replacement, and append them to the new dataset until the preferred ratio or proportion is achieved.
4. Utilize the new dataset for training or testing the ML or data mining method.

**Undersampling.**  Undersampling is a technique for balancing the class distribution of a dataset by reducing the total number of examples in the majority class. It is often used for imbalanced classification problems, where one class has significantly more examples than another class. As depicted in Fig 14, the steps of undersampling are as follows:

1. Identify the target variable and the class labels. Usually, the target variable is a binary or categorical variable, and the class labels are the possible values of the target variable.
2. Determine the desired ratio or proportion of the minority class compared to the majority class. This can be based on criteria such as achieving a balanced or near-balanced distribution, or optimizing a performance metric.
3. Randomly select a subset of examples from the majority class, with or without replacement, and remove them from the dataset. The number of examples to remove depends on the desired ratio or proportion.
4. Use the resulting dataset for training or testing an ML method.

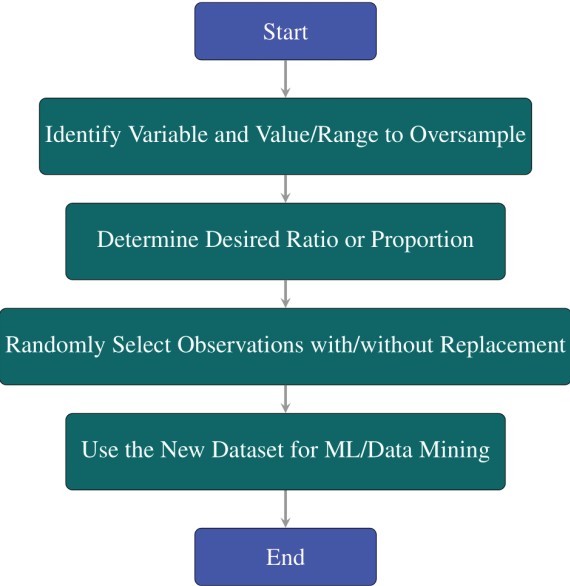

**Fig 13. Flowchart of oversampling process.**

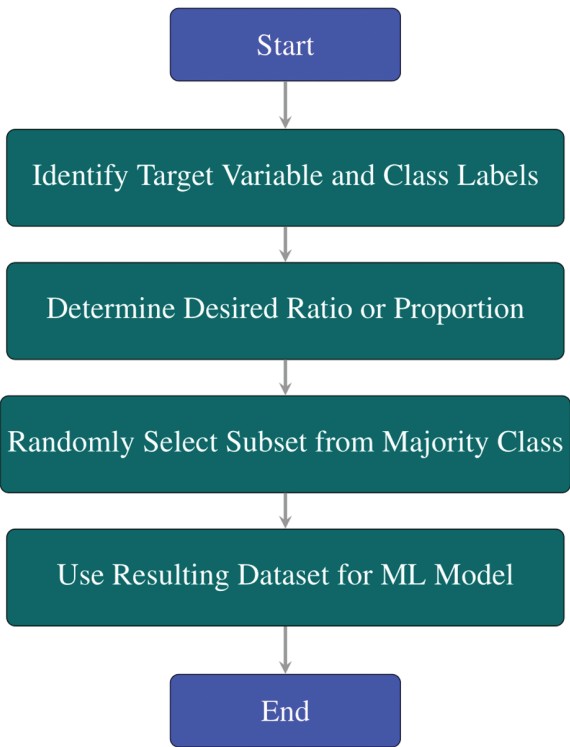

**Fig 14. Flowchart of undersampling process.**

## 4 Results and discussions

### Method evaluation

There were two main phases to the evaluation of the created methods: method fitting and method performance computation. A confusion matrix that depicts the four possible outcomes—true positive, true negative, false positive, and false negative—was used to evaluate each method's performance. In addition to the confusion metric, various metrics were calculated to compare the performance of the developed classifiers. These metrics include accuracy, cross-validation (five-fold), precision, recall, selectivity, F-measure (F1 score), balanced accuracy, G-mean, and error rate.

Accuracy is frequently believed to be the best statistic for evaluating ML techniques; however, it functions at its peak when the number of instances of each class is equal. In the case of datasets that are not evenly distributed, additional metrics besides accuracy should be utilized.

$$\text{Accuracy} = \frac{\text{number of correct predictions}}{\text{total number of predictions}} \tag{16}$$

or, equivalently,

$$\text{Accuracy} = \frac{\text{TP} + \text{TN}}{\text{TP} + \text{TN} + \text{FP} + \text{FN}} \tag{17}$$

where TP is the number of true positives, TN is the number of true negatives, FP is the number of false positives, and FN is the number of false negatives.

G-mean, or geometric mean, is a metric that considers both sensitivity and specificity, making it particularly useful for imbalanced datasets. The equation of G-mean for a set of $n$ numbers is:

$$\text{G-mean} = \sqrt[n]{x_1 \times x_2 \times \cdots \times x_n} \tag{18}$$

or, equivalently,

$$\text{G-mean} = \left( x_1 \times x_2 \times \cdots \times x_n \right)^{1/n} \tag{19}$$

where $x_1, x_2, \ldots, x_n$ are the numbers in the set.

Precision can be defined as the ratio of accurately detected positive occurrences (true positives) to the total number of instances anticipated as positive. The equation of precision is:

$$\text{Precision} = \frac{\text{number of true positives}}{\text{number of positive predictions}} \tag{20}$$

or, equivalently,

$$\text{Precision} = \frac{\text{TP}}{\text{TP} + \text{FP}} \tag{21}$$

Recall, in the context of classification models, is defined as the ratio of correctly identified positive instances to the overall number of positive occurrences in the dataset. The formula

for recall is:

$$\text{Recall} = \frac{\text{number of true positives}}{\text{number of actual positives}} \quad (22)$$

or, equivalently,

$$\text{Recall} = \frac{\text{TP}}{\text{TP} + \text{FN}} \quad (23)$$

F1-score is a measure of a method's accuracy that takes into account both precision and recall. It is the harmonic mean of precision and recall. The formula for the F-measure is:

$$\text{F-measure} = \frac{2 \times \text{Precision} \times \text{Recall}}{\text{Precision} + \text{Recall}} \quad (24)$$

or, equivalently,

$$\text{F-measure} = \frac{2 \times \text{TP}}{2 \times \text{TP} + \text{FP} + \text{FN}} \quad (25)$$

Balanced accuracy, which is the average of sensitivity and specificity, is another metric that is useful for imbalanced datasets as it gives equal weight to both classes. The formula commonly used to calculate balanced accuracy is defined as the arithmetic mean of the true positive ratio and the true negative ratio, divided by two.

$$\text{Balanced Accuracy} = \frac{\text{Sensitivity} + \text{Specificity}}{2} \quad (26)$$

or, equivalently,

$$\text{Balanced Accuracy} = \frac{\text{TPR} + \text{TNR}}{2} \quad (27)$$

where TPR is the true positive rate, and TNR is the true negative rate.

Error rate, the proportion of incorrect predictions, is a straightforward and intuitive metric that can be used alongside other metrics to provide a comprehensive view of method performance. The formula for error rate is:

$$\text{Error Rate} = \frac{\text{number of incorrect predictions}}{\text{total number of predictions}} \quad (28)$$

or, equivalently,

$$\text{Error Rate} = 1 - \text{Accuracy} \quad (29)$$

Selectivity is the proportion of true negatives (correctly identified negative instances) over the total number of instances predicted as negative. The formula for Selectivity is:

$$\text{Selectivity} = \frac{\text{TN}}{\text{TN} + \text{FP}} \quad (30)$$

or, equivalently,

$$\text{Selectivity} = \frac{\text{TN}}{\text{number of actual negatives}} \tag{31}$$

Cross-validation, particularly the five-fold cross-validation technique, is a resampling methodology employed to assess ML systems using a limited dataset. The process encompasses a solitary parameter denoted as $k$, which pertains to the quantity of groups into which a specific data sample is divided. In the context of five-fold cross-validation, the dataset is partitioned into five distinct groups. Then, the method is trained and tested five times, providing a more robust estimate of method performance.

Lastly, Standard Deviation measures the amount of variation or dispersion in a set of values. In the context of machine learning, it can be used to evaluate the spread of prediction errors (residuals). A lower standard deviation indicates that the predictions are closer to the mean prediction, suggesting a more consistent model.

The standard deviation ($\sigma$) is calculated using the following formula:

$$\sigma = \sqrt{\frac{1}{N}\sum_{i=1}^{N}(x_i - \mu)^2} \tag{32}$$

where:

- $N$ is the number of predictions.
- $x_i$ represents each individual prediction.
- $\mu$ is the mean of the predictions.

## Comparison of evaluation metrics for bagging ML methods

This section compares evaluation metrics for different bagging machine learning methods. Different metrics, including accuracy, precision, recall, and F1-score, are used to provide a comprehensive assessment of each model's performance.

**Evaluation metrics for bagging ML methods (For Oversampled Data).** The Table 2 presents performance metrics for six different bagging models: kNN, LR, DT, NB, SVM, and RF for oversampled data. The oversampled data table shows that Bagging-kNN has the highest accuracy at 98.37% and the lowest error rate at 1.63%, indicating its robustness. Bagging-LR and Bagging-NB also perform well, with accuracies of 97.56% and 96.95%, respectively. Bagging-DT has the lowest accuracy at 95.12%, despite a relatively high recall at 94.83%. Bagging-SVM and Bagging-RF have similar accuracies around 95.9%, displaying moderate strengths across metrics.

The top chart in Fig 15 shows the performance of six machine learning models across various metrics. The bottom chart in Fig 14 shows the standard deviation of the performance metrics. Overall, k-NN outperformed other models with 98.37% accuracy and 0.192% standard deviation. Logistic Regression and Naive Bayes also performed well with accuracies of 97.56% and 96.95%, but with higher variabilities of 0.533% and 0.445%. Decision Tree, despite achieving 95.12% accuracy, had the highest standard deviation at 0.755%, suggesting less reliability. Random Forest and SVM provided a balance between performance and consistency, with SVM at 95.93% accuracy and 0.441% standard deviation, and Random Forest at 95.93% accuracy and 0.319% standard deviation.

**Table 2. Comparison of bagging ML methods for oversampled dataset.**

| Model | Accuracy | CV | Precision | Recall | Selectivity | F1-score | Balanced Accuracy | G-mean | Error- rate | AUC | SD | processing time (s) |
|---|---|---|---|---|---|---|---|---|---|---|---|---|
| Bagging-kNN | 98.37 | 98.23 | 97.67 | 97.93 | 98.18 | 97.79 | 98.06 | 98.05 | 1.63 | 0.98 | 0.192 | 582 |
| Bagging-LR | 97.56 | 97.23 | 96.76 | 95.89 | 96.93 | 96.32 | 96.41 | 96.41 | 2.44 | 0.96 | 0.533 | 602 |
| Bagging-DT | 95.12 | 94.56 | 93.69 | 94.83 | 95.03 | 94.26 | 94.93 | 94.93 | 4.88 | 0.95 | 0.755 | 614 |
| Bagging-NB | 96.95 | 95.89 | 95.36 | 95.66 | 96.56 | 95.51 | 96.11 | 96.11 | 3.05 | 0.96 | 0.445 | 592 |
| Bagging-SVM | 95.93 | 95.78 | 94.65 | 94.88 | 95.69 | 94.76 | 95.29 | 95.28 | 4.07 | 0.95 | 0.441 | 665 |
| Bagging-RF | 95.89 | 95.33 | 94.89 | 94.73 | 95.56 | 94.81 | 95.15 | 95.14 | 4.07 | 0.95 | 0.319 | 612 |

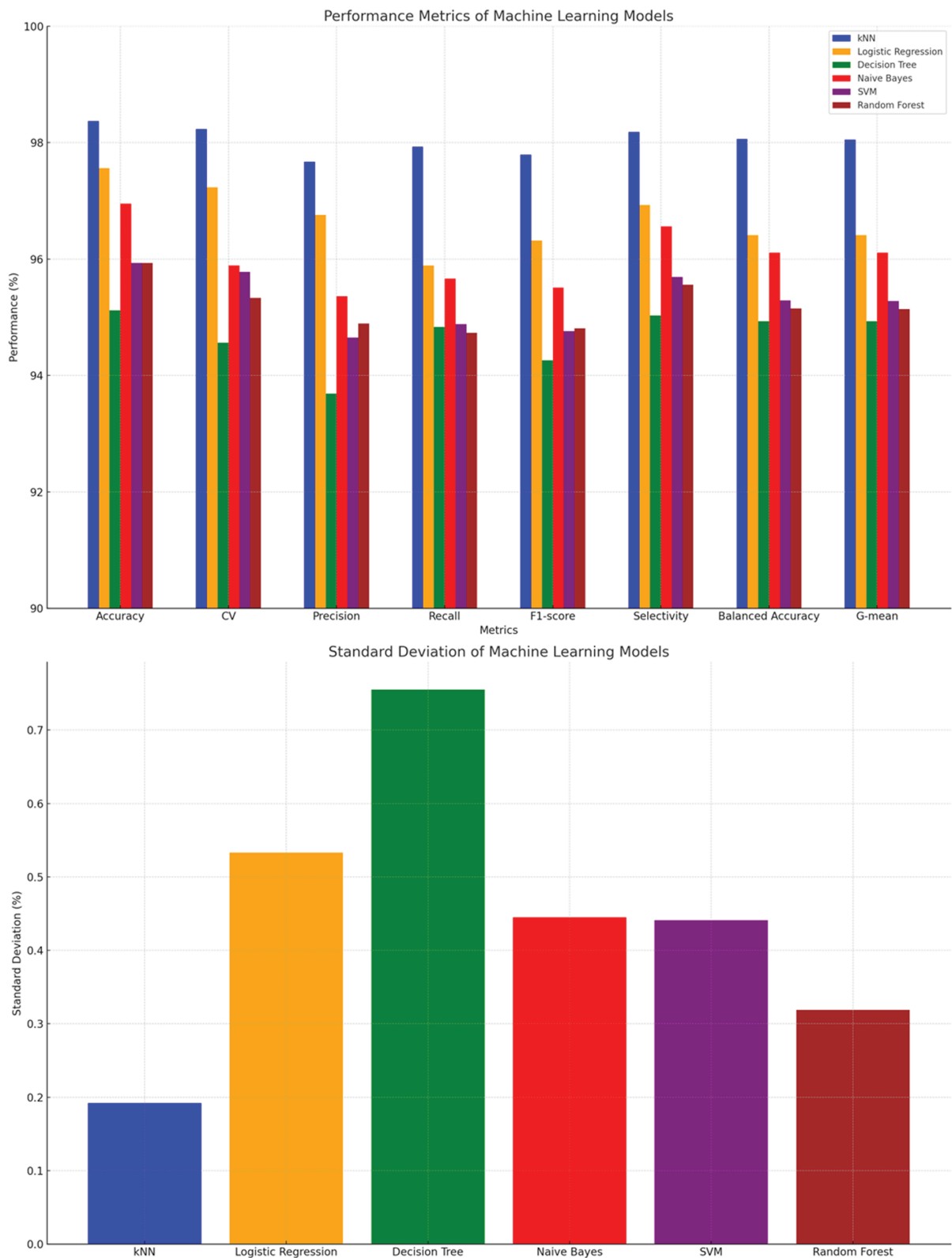

**Fig 15. Comparison of evaluation metrics for bagging-ML methods (For Oversampled Data).**

Confusion Matrices for Different Methods (Oversampling)

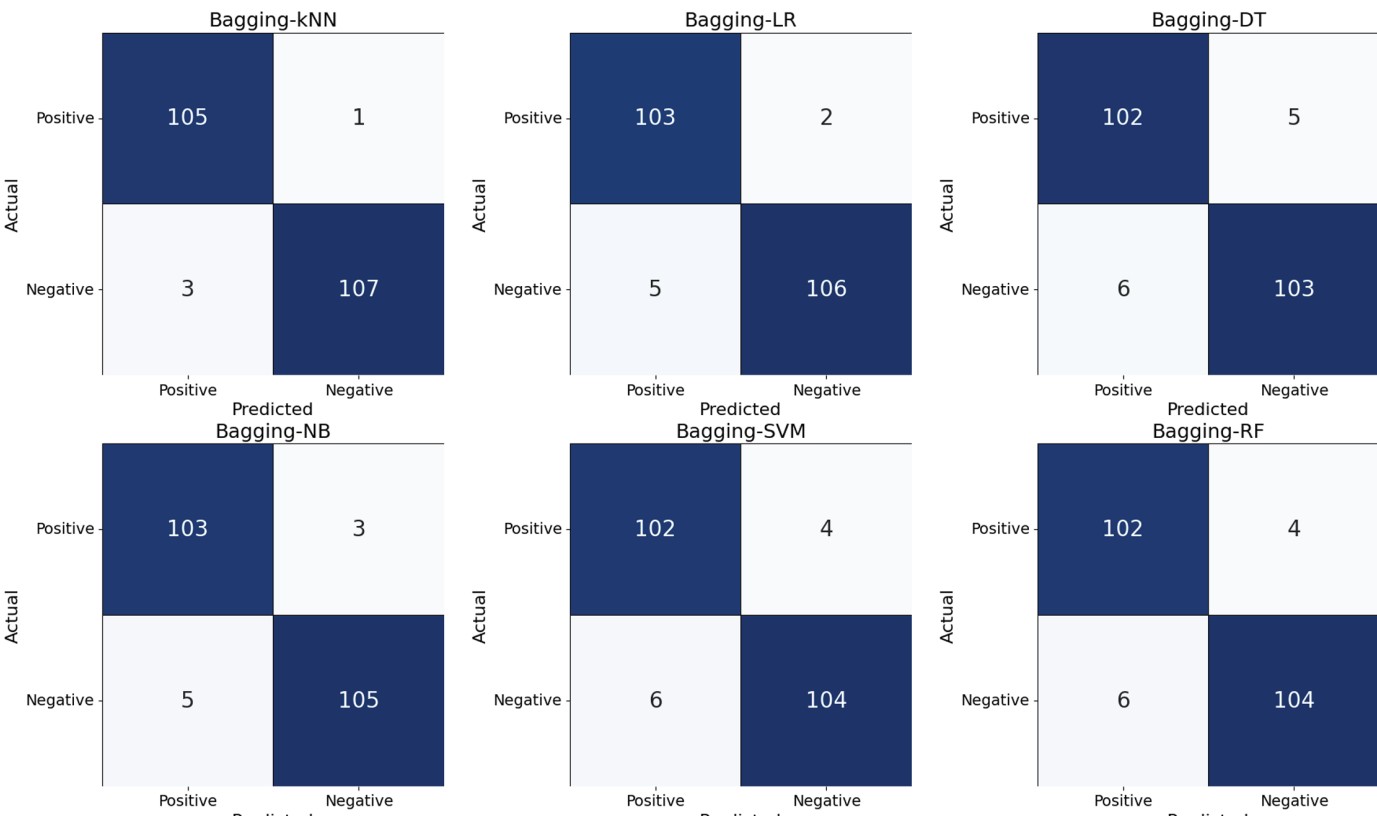

**Fig 16. Confusion matrices for bagging ML methods (For Oversampled Dataset).**

The confusion matrixes in Fig 16 visualization display the performance of six different bagging-enhanced machine learning models—Bagging-kNN, Bagging-LR, Bagging-DT, Bagging-NB, Bagging-SVM, and Bagging-RF—on predicting potential hepatitis C virus patients using an oversampled dataset. Bagging-kNN emerges as the most effective model with the lowest number of misclassifications. Models like Bagging-DT, Bagging-SVM, and Bagging-RF show potential but require further refinement to reduce false negatives.

As depicted in Fig 17, the ROC (Receiver Operating Characteristic) curves and AUC (Area Under the Curve) values for different bagging machine learning models applied to an oversampled dataset demonstrate varying levels of performance. Bagging-kNN shows the highest effectiveness with an AUC of 0.98, indicating its superior ability to distinguish between positive and negative classes, characterized by a sharp rise towards the top-left corner of the ROC curve. Bagging-LR and Bagging-NB both exhibit strong performance with AUCs of 0.96, suggesting a good balance between sensitivity and specificity, though slightly less optimal than Bagging-kNN. Bagging-DT, Bagging-SVM, and Bagging-RF each have AUCs of 0.95, indicating they are also effective but slightly less optimal compared to Bagging-kNN and Bagging-LR. Overall, Bagging-kNN emerges as the most effective model, followed closely by Bagging-LR and Bagging-NB.

**Evaluation metrics for bagging ML methods (For Undersampled Data).** The Table 3 presents performance metrics for six different bagging models: kNN, LR, DT, NB, SVM,

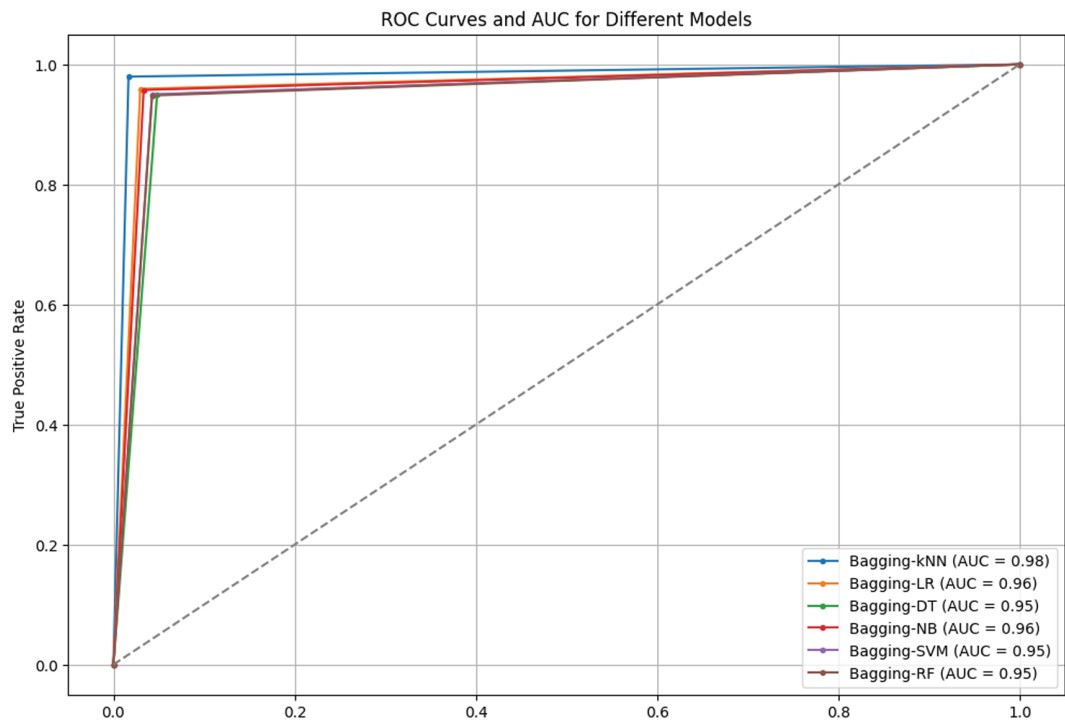

**Fig 17. AUC for bagging ML methods (For Oversampled Dataset).**

and RF for undersampled data. In the undersampled data table, Bagging-LR achieves the highest accuracy at 97.56%, with strong recall at 96.64% and F1-score at 96.73%. Bagging-kNN and Bagging-RF also perform well, with accuracies of 96.38% and 96.48%, respectively. Bagging-DT has the lowest accuracy at 93.96%, with an error rate of 6.04%. Bagging-SVM and Bagging-NB show accuracies of 95.48% and 94.58%, respectively, demonstrating moderate performance.

The top chart in Fig 18 shows the performance of six machine learning models across various metrics. The bottom chart in Fig 18 shows the standard deviation of the performance metrics. Overall, Logistic Regression outperformed other models with a 97.56% accuracy and a standard deviation of 0.382%. Random Forest followed closely with a 96.48% accuracy but had the highest variability with a standard deviation of 0.724%. kNN also performed well with a 96.38% accuracy and a standard deviation of 0.444%. SVM showed balanced performance with a 95.48% accuracy and the lowest standard deviation at 0.265%. Naive Bayes and Decision Tree had lower accuracies at 94.58% and 93.96%, with standard deviations of 0.408% and 0.315%, respectively. These results indicate that Logistic Regression and SVM are suitable choices for tasks involving undersampled datasets, with Logistic Regression excelling in performance and SVM in consistency.

The confusion matrixes in Fig 19 visualization display the performance of six different bagging-enhanced machine learning models—Bagging-kNN, Bagging-LR, Bagging-DT, Bagging-NB, Bagging-SVM, and Bagging-RF—on predicting potential hepatitis C virus patients using an under-sampled dataset. Bagging-LR stands out as the most accurate and reliable model for undersampled data, with the lowest number of misclassifications. Bagging-NB and Bagging-DT show the highest need for improvement due to their high false positive and

**Table 3. Comparison of bagging ML methods for undersampled dataset.**

| Model | Accuracy | CV | Precision | Recall | Selectivity | F1-score | Balanced Accuracy | G-mean | Error- rate | AUC | SD | processing time (s) |
|---|---|---|---|---|---|---|---|---|---|---|---|---|
| Bagging-KNN | 96.38 | 96.12 | 95.79 | 95.36 | 95.81 | 95.57 | 95.59 | 95.59 | 3.62 | 0.92 | 0.444 | 609 |
| Bagging-LR | **97.56** | **97.23** | **96.83** | **96.64** | **96.75** | **96.73** | **96.69** | **96.70** | **2.44** | **0.94** | 0.382 | 634 |
| Bagging-SVM | 95.48 | 95.31 | 94.86 | 94.39 | 94.72 | 94.62 | 94.55 | 94.55 | 4.52 | 0.90 | **0.265** | 623 |
| Bagging-NB | 94.58 | 94.29 | 92.49 | 92.26 | 92.72 | 92.37 | 92.49 | 92.49 | 5.42 | 0.89 | 0.408 | 612 |
| Bagging-DT | 93.96 | 93.64 | 92.82 | 92.86 | 92.48 | 92.84 | 92.67 | 92.67 | 6.04 | 0.83 | 0.315 | **607** |
| Bagging-RF | 96.48 | 96.17 | 95.20 | 95.72 | 95.83 | 95.46 | 95.77 | 95.77 | 3.52 | 0.92 | 0.724 | 620 |

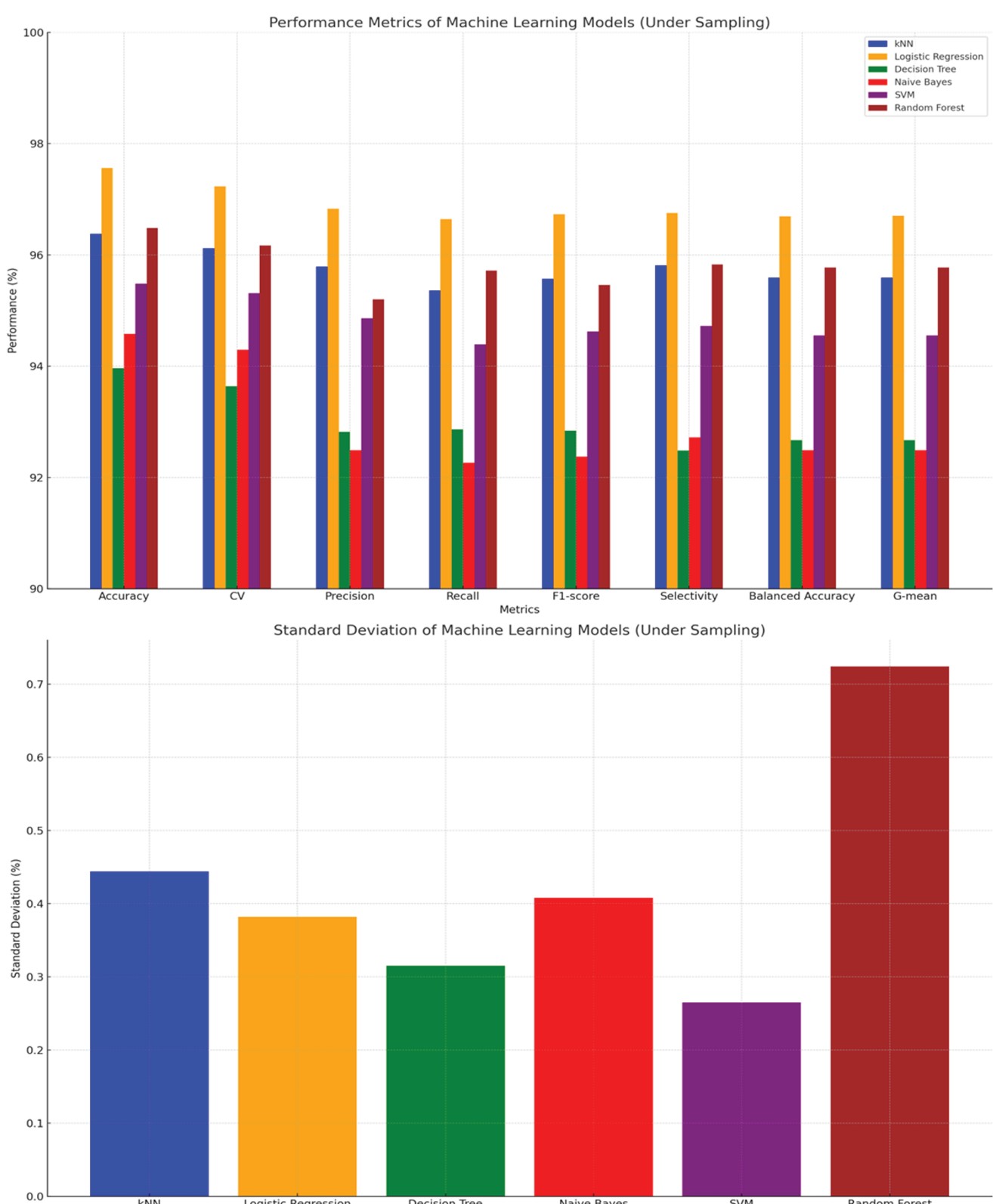

**Fig 18. Comparison of evaluation metrics for bagging-ML methods (For Undersampled Data).**

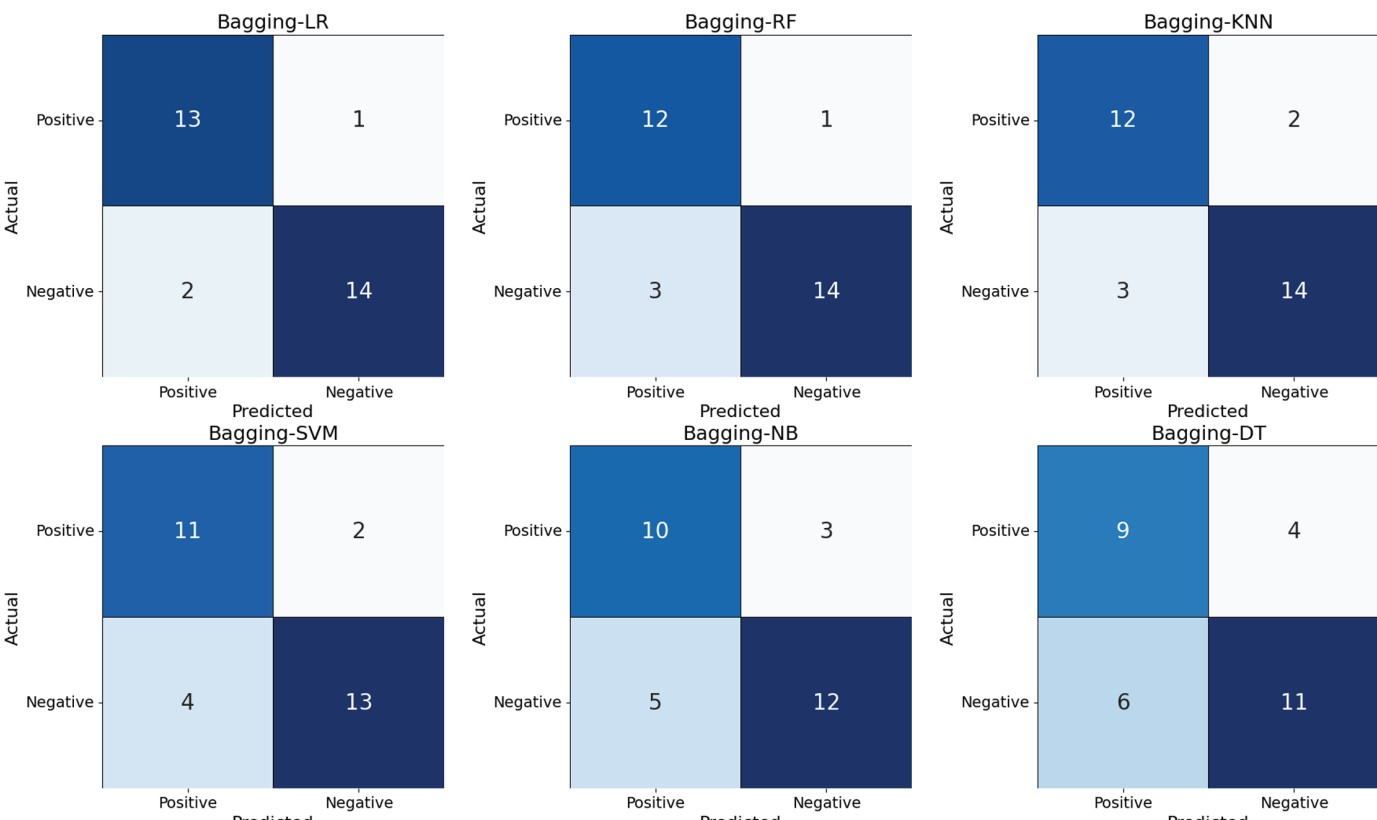

**Fig 19. Confusion matrices for bagging ML methods (For Undersampled Dataset).**

negative rates. These models may not be suitable for immediate clinical application without significant enhancement.

As visible in Fig 20, the ROC curves and AUC values for different bagging machine learning models applied to an undersampled dataset highlight the effectiveness of each model in distinguishing between positive and negative classes. Bagging-LR demonstrates the highest performance with an AUC of 0.94, showing an excellent balance between sensitivity and specificity. Bagging-kNN and Bagging-RF both perform strongly with AUCs of 0.92, indicating high capabilities to differentiate between classes, characterized by steep rises towards the top-left corner of their ROC curves. Bagging-NB and Bagging-SVM also show good performance with AUCs of 0.89 and 0.90 respectively, suggesting they are effective but slightly less optimal than Bagging-LR. Bagging-DT, with an AUC of 0.83, shows moderate performance and is less effective compared to the other models. Overall, Bagging-LR is identified as the most effective model for the undersampled dataset, followed closely by Bagging-kNN and Bagging-RF, making all these models, except Bagging-DT, viable options for clinical predictions with undersampled data.

**Comparison of accuracy for bagging-ML methods in this study.** The Fig 21 visually differentiates between the two sampling techniques—oversampling is represented by non-checkered bars while undersampling is depicted by checkered bars. This facilitates easy comparison between the methods' performance under each method. The bars are arranged in

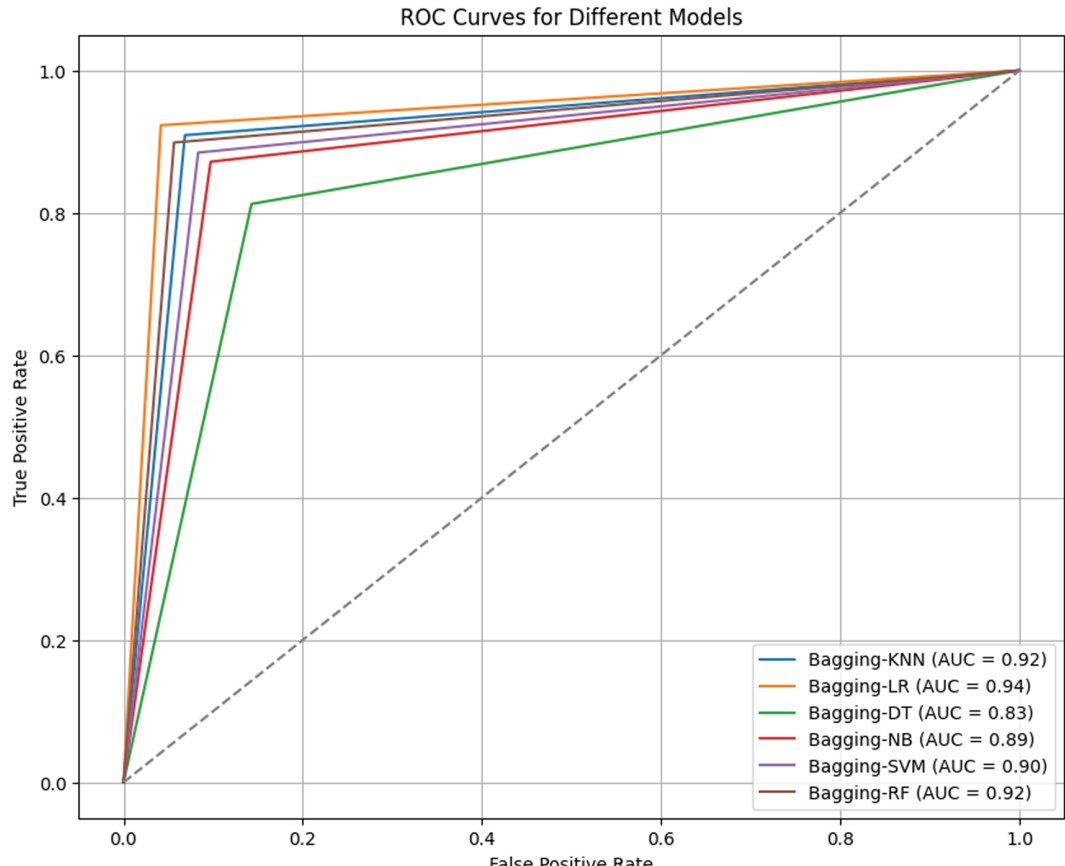

**Fig 20. AUC for bagging ML methods (For Undersampled Dataset).**

descending order of accuracy from left to right, allowing quick identification of the best and worst performers. Overall, Bagging exhibits high accuracy across methods, with values ranging from 93.96% to 98.37%. The top three most accurate methods under oversampling are Bagging k-NN, Bagging LR, and Bagging RF. This highlights these methods benefit from oversampling. The undersampling methods mostly show slightly lower accuracy than their oversampled counterparts. The difference is most prominent in Bagging SVM, suggesting undersampling may not be as effective for this method. Finally, Bagging k-NN (Oversampling) achieves the highest accuracy while Bagging SVM (Undersampling) has the lowest accuracy out of all the presented methods.

An ensemble method with Bagging was used for the methods (SVM, k-NN, LR, RF, NB, and DT) to enhance their performance. This approach combines multiple methods to produce a final predictive method that performs better than any individual method. Following the enhancement of the methods (SVM, k-NN, LR, RF, NB, and DT), they were trained on the previously assembled dataset. Then, predefined measures were used to judge how well these newly created classification methods worked. The total number of data from the testing set was used to figure out how well each method worked in the end. At first, a confusion measure was made for each classifier. Subsequently, several metrics including accuracy, recall (sensitivity), precision, F1 score, g-mean, balanced accuracy, error rate, standard deviation and results from five-fold cross validation were calculated for each method based on the confusion

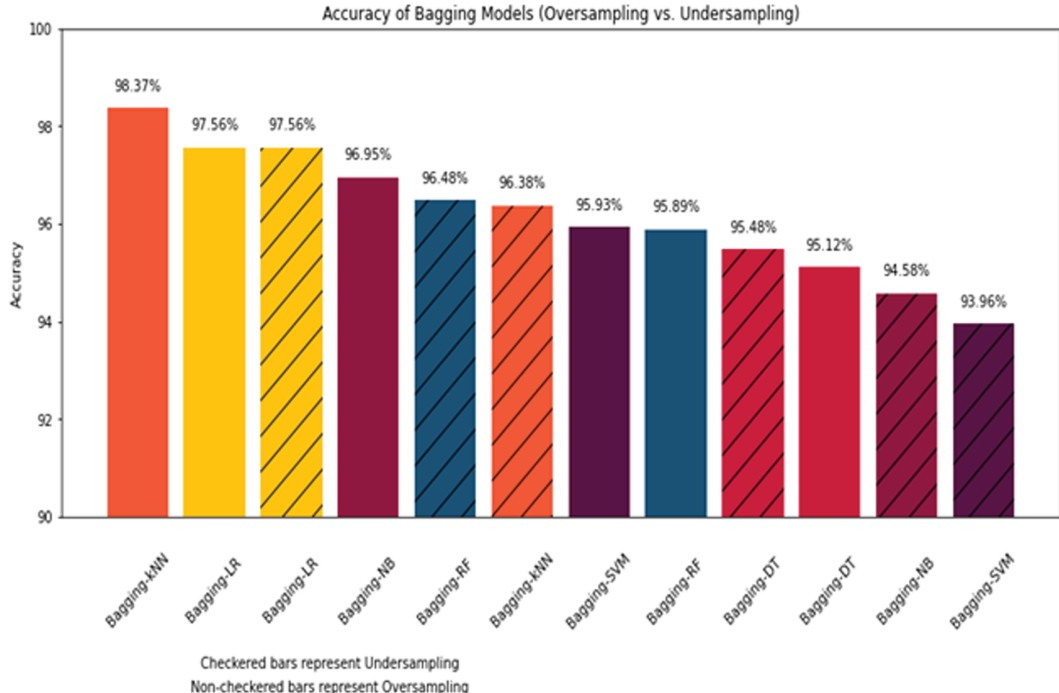

**Fig 21. Comparison of accuracy for bagging-ML methods used in this study.**

metric. The comparison of these method metrics is presented in. In the process of selecting the best-performing method to propose for use, the focus was on the independent performance of each individual method. Based on the assessment metrics and additional data, it can be concluded that the Bagging k-NN classifier demonstrated superior performance compared to other classifiers, as indicated by the confusion measure of the alternative approaches. The RF classifier demonstrated a superior level of accuracy, reaching 98.37%, which surpassed the performance of all other classifiers. In terms of g-mean, f1 score, balanced accuracy, precision, recall, error rate, and results from five-fold cross validation, the k-NN method also demonstrated superior performance among all the other algorithms when trained on oversampled dataset, underscoring its effectiveness in classifying patients.

## Discussion

As already mentioned in previous sections of this paper, the ability to classify patients' health status swiftly and accurately is of utmost importance for healthcare providers. Therefore, in this study, A diverse range of ML approaches were employed in order to categorize patients who were suspected of being infected with the HCV. Additional advancements were made in the refinement of methodologies for determining the stage of HCV infection by utilizing data that has been subjected to training. The evaluation of the classifiers' performance and accuracy was conducted. The success of this research work with high accuracy proves to justify the rising popularity of ML classification methods among health specialists. Table 4 provides a detailed view on the comparison of accuracies among existing works and that of this paper.

Six ML methods were employed in this endeavor, namely SVM, RF, LR, K Nearest Neighbor (k-NN), NB and DT. All these methods were enhanced by integrating them with bagging (bootstrap aggregation), which improved their overall stability and accuracy. This novel and                     AQ2

**Table 4. Highest accuracy of ML methods compared to previous works.**

| References | Predicted | Models | Methods | Metric | Results |
|---|---|---|---|---|---|
| [26] | HCV | Neural Network | ML techniques | accuracy | 95.12% |
| [20] | Cirrhosis | Bayesian Nets | ML techniques | accuracy | 68.9% |
| [21] | HCV | RF | ML techniques, Data preprocessing, Feature selection | accuracy | 54.56% |
| [27] | HCV | SVM fine Gaussian | ML techniques | accuracy | 97.9% |
| [28] | HCV | LR, SVM | ML techniques | accuracy | 95% |
| [36] | Cirrhosis, Fibrosis | RF, DT | Data preprocessing, Feature extraction, ML techniques | accuracy | 96.32%, 98.11% |
| [33] | HCV | LR, NB, SVM, k-NN, DT, RF | Data preprocessing, Feature extraction, ML techniques | accuracy | 95.67%, 92.43%, 94.59%, 94.57%, 96.75%, 97.29% |
| Our Study | HCV | Bagging k-NN | Undersampling, Feature reduction, Ensemble ML techniques | accuracy | 96.38% |
| | | Bagging LR | | | 97.56% |
| | | Bagging NB | | | 94.58% |
| | | Bagging RF | | | 96.48% |
| | | Bagging SVM | | | 95.93% |
| | | Bagging DT | | | 93.96% |
| Our Study | HCV | Bagging k-NN | Oversampling, Feature reduction, Ensembling ML techniques | accuracy | **98.37%** |
| | | Bagging LR | | | 97.56% |
| | | Bagging NB | | | 96.95% |
| | | Bagging RF | | | 95.89% |
| | | Bagging SVM | | | 95.93% |
| | | Bagging DT | | | 95.12% |

original approach contributes to the improvement of healthcare diagnostics by allowing for more precise classification of suspected HCV infections in patients. This study emphasized the importance of data preprocessing to increase the accuracy and performance of the methods used. Missing data in the dataset were addressed by replacing them with the MODE values, which is a robust method that minimizes the impact of these missing values on the classification results. Feature reductions were done using one-way ANOVA-F statistic testing, and redundant variables were discarded, thus optimizing the data for ML. To counter the issue of class imbalance in the dataset Lichtinghagen et al., 2020), undersampling and oversampling techniques were employed, thereby enhancing the performance of the classifiers. Furthermore, various method evaluation metrics such as accuracy, cross validation (five-fold), precision, recall, selectivity, f1-score, balanced accuracy, g-mean, standard deviation and error rate were used to evaluate the performance of the employed ML methods. The preprocessing techniques, coupled with robust method evaluation techniques set this study apart from other research and substantiate the remarkable accuracy of this work's classification methods. It is noteworthy that the best performing method in this study was the Bagging k-NN method, which achieved highest accuracy of 98.37%. This surpasses the accuracy level obtained in previous research, signifying the effectiveness of this work's approach.

In essence, this study makes several key contributions that advance the field of HCV infection prediction using ML techniques. First, this research pioneered an innovative approach to handling missing values in the dataset. Rather than using typical imputation techniques like mean or median substitution, this study substituted missing values with mode values. This allowed for preserving the inherent reliability of the data. Second, addressing class imbalance

was done using under-sampling and over-sampling methods rather than the commonly used SMOTE technique. The former approaches maintained the natural data distribution while balancing classes, proving more effective. The third contribution was the application of one-way ANOVA F-statistic testing for feature reduction. This robust statistical method enabled superior feature selection compared to typical feature extraction methods, improving method performance. Fourth, this study uniquely integrated Bagging with multiple ML methods like k-NN, RF, LR, DT, SVM, and NB. Bagging ensemble enhanced method stability and prediction capability significantly. Fifth, a comparative analysis of ML methods from earlier studies was done in comparison with this study's outcomes. Most significantly, the sixth contribution achieved the highest known accuracy in HCV prediction 98.37% using Bagging k-NN. This unprecedented accuracy establishes a new benchmark in this research field.Despite its contributions, this study has several limitations that warrant further research. The dataset may not be fully representative of diverse populations, requiring validation on broader demographics. ANOVA F-test captures only linear relationships; advanced feature selection methods could enhance interpretability. Addressing evolving class imbalances through dynamic resampling or cost-sensitive learning is crucial. Exploring alternative ensemble techniques like Boosting and Stacking may further improve performance.

## 5 Conclusion

Data science, with its multidisciplinary approach, plays a crucial role in enhancing medical decision-making, particularly in diagnosing and treating diseases like Hepatitis C Virus (HCV), a global health concern. Early and accurate diagnosis of HCV is essential for effective treatment, and this study advances that goal by integrating data science methodologies with visualization techniques to predict HCV more efficiently. Unlike traditional diagnostic models, this approach goes beyond basic statistical learning by employing sophisticated classification and feature reduction techniques to improve predictive accuracy. A key innovation of this study is the use of ensemble-based oversampling and undersampling techniques, instead of the widely used SMOTE method, to address class imbalance. This method, combined with Bagging-enhanced machine learning models—such as SVM, k-NN, LR, RF, NB, and DT—significantly improved classification performance. The Bagging k-NN method achieved an accuracy of 98.37% under oversampling conditions, while Bagging LR reached 97.56% under undersampling conditions, setting a new benchmark for HCV prediction. Furthermore, this study identified significant feature correlations, enabling the development of an efficient decision support system for HCV diagnosis. To implement this model in real-world clinical settings, several key resources and steps are required. First, integration with Electronic Health Records (EHRs) is essential to enable real-time data analysis and automated HCV risk assessment. This can be facilitated through APIs or cloud-based services that allow seamless data transfer between clinical databases and the prediction model. Additionally, a hospital-grade server or cloud computing infrastructure is necessary to run ML models efficiently, while edge computing devices could support real-time inference in point-of-care settings. To make the system accessible to healthcare professionals, a user-friendly graphical interface should be developed, displaying risk predictions and patient classifications in an intuitive format. This interface should include clear visualizations and explanations of model predictions to assist clinical decision-making. Future research will focus on integrating deep learning architectures to improve predictive accuracy and incorporating multi-modal data sources, such as liver imaging and genetic markers, to enhance diagnostic precision. Additionally, real-time patient monitoring and adaptive learning techniques could further optimize model performance. By addressing these aspects, this study contributes to the development of a clinically

viable, AI-powered decision support system, ultimately aiding early detection and personalized treatment strategies for HCV patients.

## Author contributions

**Conceptualization:** Ekramul Haque Tusher.

**Formal analysis:** Ekramul Haque Tusher, Abdullah Akib.

**Funding acquisition:** Mohd Arfian Ismail, Lubna A.Gabralla, Ashraf Osman Ibrahim, Hafizan Mat Som.

**Investigation:** Ekramul Haque Tusher, Mohd Arfian Ismail.

**Methodology:** Ekramul Haque Tusher.

**Supervision:** Mohd Arfian Ismail.

**Visualization:** Ekramul Haque Tusher, Abdullah Akib.

**Writing – original draft:** Ekramul Haque Tusher.

**Writing – review & editing:** Mohd Arfian Ismail, Abdullah Akib, Lubna A.Gabralla, Hafizan Mat Som, Muhammad Akmal Remli.

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

# AUTHOR QUERY FORM

| AQ1. | Figure 8, 10, 11, 12,15, 18 & 21 – The quality of the image is poor and pixelated. Hence please supply a corrected version with an unpixelated typeface. |
|------|------|
| AQ2. | Please note that the References [39, 43] are cited in the text not in the reference list. Kindly check. |