## [Decision Letter · Decision Letter 0]

17 Feb 2025

PONE-D-25-00479Comparative Investigation of Bagging Enhanced Machine Learning for Early Detection of HCV Infections Using Class Imbalance Technique with Feature SelectionPLOS ONE

Dear Dr. Ismail,

Thank you for submitting your manuscript to PLOS ONE. After careful consideration, we feel that it has merit but does not fully meet PLOS ONE’s publication criteria as it currently stands. Therefore, we invite you to submit a revised version of the manuscript that addresses the points raised during the review process.

**ACADEMIC EDITOR: Major revisions**

We look forward to receiving your revised manuscript.

Kind regards,

Agbotiname Lucky Imoize

Academic Editor

PLOS ONE

3. Thank you for stating the following financial disclosure:  [This study was supported by Fundamental Research Grant (FRGS) withFRGS/1/2022/ICT02/UMP/02/2 (RDU220134) from the Ministry of Higher Education 937 Malaysia and the authors extend their appreciation to the Deanship of Research and

Graduate Studies at King Khalid University for funding this work through Large Research Project under grant number RGP2/319/45.].  Please state what role the funders took in the study.  If the funders had no role, please state: "The funders had no role in study design, data collection and analysis, decision to publish, or preparation of the manuscript." If this statement is not correct you must amend it as needed.

4. Thank you for uploading your study's underlying data set. Unfortunately, the repository you have noted in your Data Availability statement does not qualify as an acceptable data repository according to PLOS's standards.

At this time, please upload the minimal data set necessary to replicate your study's findings to a stable, public repository (such as figshare or Dryad) and provide us with the relevant URLs, DOIs, or accession numbers that may be used to access these data. For a list of recommended repositories and additional information on PLOS standards for data deposition, please see https://journals.plos.org/plosone/s/recommended-repositories

Additional Editor Comments:

The implementation is lacking in clarity. The authors must rework this section and give more clarity on the materials and methods.

In addition, address the reviewers' completely.

Reviewers' comments:

Reviewer's Responses to Questions

**Comments to the Author**

1. Is the manuscript technically sound, and do the data support the conclusions?

Reviewer #1: Yes

Reviewer #2: Yes

2. Has the statistical analysis been performed appropriately and rigorously? 

Reviewer #1: Yes

Reviewer #2: N/A

3. Have the authors made all data underlying the findings in their manuscript fully available?

Reviewer #1: No

Reviewer #2: No

4. Is the manuscript presented in an intelligible fashion and written in standard English?

Reviewer #1: Yes

Reviewer #2: Yes

5. Review Comments to the Author

Reviewer #1: Regarding the manuscript titled "Comparative Investigation of Bagging Enhanced Machine Learning for Early Detection of HCV Infections Using Class Imbalance Technique with Feature Selection"

I reviewed this manuscript. I found that this manuscript uses an acceptable methodology. Details are provided in the presentation of the material. . The research questions are well answered. It seems that with minimal revisions it can be recommended for publication. Therefore, I draw attention to the following points.

1: The abstract is well written. However, the introduction seems to suffer from the lack of one topic, and that is a brief definition of artificial intelligence and machine learning and its applications in the field of healthcare. For this section, you can refer to more studies and applications of machine learning in various fields of health. You can also use the following studies.

"Artificial intelligence in drug discovery and development against antimicrobial resistance: A narrative review"

"Mobile apps for COVID-19 detection and diagnosis for future pandemic control: Multidimensional systematic review"

Using these studies can provide important resources of machine learning in the field of healthcare.

2: Regarding the choice of these algorithms, you need to explain why you used the bugging algorithms at all?

The rest of the issues seem to be well written and presented clearly and do not need any special changes and can be accepted as is.

Reviewer #2: 1. The paper lacks a clear explanation of why certain features were selected or removed based on the ANOVA F-test results, especially regarding the clinical relevance.

2. The methodology section lacks detail about the specific parameters used for each machine learning algorithm.

3. There is no discussion about the computational complexity or processing time of the different methods.

4. There is no validation using an external dataset to confirm the robustness of the proposed methods.

5. The authors do not justify why they chose a 80-20 split for training-testing instead of other common ratios.

6. The figures showing confusion matrices (14 and 17) are too small and difficult to interpret.

7. The paper does not address the potential bias in the dataset regarding demographic factors like age and gender distribution.

8. The discussion section does not adequately address the limitations of the study or potential areas for future research.

9. The paper lacks a clear explanation of how the proposed method could be implemented in real-world clinical settings and what resources would be required.

6. PLOS authors have the option to publish the peer review history of their article (what does this mean?). If published, this will include your full peer review and any attached files.

Reviewer #1: **Yes: **Mustafa Ghaderzadeh

Reviewer #2: No

---

## [Author Response · Author response to Decision Letter 1]

6 Apr 2025

the respond to reviewers can be found in file Response_to_Reviewers_Plos_one.docx

---

## [Decision Letter · Decision Letter 1]

16 Apr 2025

PONE-D-25-00479R1Comparative Investigation of Bagging Enhanced Machine Learning for Early Detection of HCV Infections Using Class Imbalance Technique with Feature SelectionPLOS ONE

Dear Dr. Ismail,

Thank you for submitting your manuscript to PLOS ONE. After careful consideration, we feel that it has merit but does not fully meet PLOS ONE’s publication criteria as it currently stands. Therefore, we invite you to submit a revised version of the manuscript that addresses the points raised during the review process.

**ACADEMIC EDITOR: Minor revisions**==============================

We look forward to receiving your revised manuscript.

Kind regards,

Agbotiname Lucky Imoize

Academic Editor

PLOS ONE

Journal Requirements:

Additional Editor Comments:

The authors should recreate the blurred flowcharts, especially figures 2-7. Others can be improved as well. I think it has to do with the tool used. I suggest that you use Visio to create excellent figures.

Additionally, minor English editing is required.

Reviewers' comments:

Reviewer's Responses to Questions

**Comments to the Author**

1. If the authors have adequately addressed your comments raised in a previous round of review and you feel that this manuscript is now acceptable for publication, you may indicate that here to bypass the “Comments to the Author” section, enter your conflict of interest statement in the “Confidential to Editor” section, and submit your "Accept" recommendation.

Reviewer #1: All comments have been addressed

Reviewer #2: (No Response)

2. Is the manuscript technically sound, and do the data support the conclusions?

Reviewer #1: Yes

Reviewer #2: (No Response)

3. Has the statistical analysis been performed appropriately and rigorously? 

Reviewer #1: Yes

Reviewer #2: (No Response)

4. Have the authors made all data underlying the findings in their manuscript fully available?

Reviewer #1: Yes

Reviewer #2: (No Response)

5. Is the manuscript presented in an intelligible fashion and written in standard English?

Reviewer #1: Yes

Reviewer #2: (No Response)

6. Review Comments to the Author

Reviewer #1: Regarding this manuscript, I would like to inform you that I have re-examined this study. I found that all revisions were correctly addressed and the quality of this study has now reached an acceptable level. Therefore, I strongly recommend it for publication.

Reviewer #2: The authors have accurately considered all the given comments; hence, the paper is worthy of acceptance.

7. PLOS authors have the option to publish the peer review history of their article (what does this mean?). If published, this will include your full peer review and any attached files.

Reviewer #1: No

Reviewer #2: No

---

## [Author Response · Author response to Decision Letter 2]

16 May 2025

Associate Editor Comment-1 : The authors should recreate the blurred flowcharts, especially figures 2-7. Others can be improved as well. I think it has to do with the tool used. I suggest that you use Visio to create excellent figures.

Additionally, minor English editing is required.

Author Response-1: Thank you for your feedback. I have redrawn the flowcharts, including figures 2-7, as well as figures 11 and 12, in response to your suggestion. I believe the updated figures should now meet the clarity standards you were expecting.

Additionally, I made minor revisions to the abstract and conclusion, as requested, to improve the language and readability.

---

## [Editor Report · Decision Letter 2]

1 Jun 2025

Comparative Investigation of Bagging Enhanced Machine Learning for Early Detection of HCV Infections Using Class Imbalance Technique with Feature Selection

PONE-D-25-00479R2

Dear Dr. Ismail,

We’re pleased to inform you that your manuscript has been judged scientifically suitable for publication and will be formally accepted for publication once it meets all outstanding technical requirements.

Kind regards,

Agbotiname Lucky Imoize

Academic Editor

PLOS ONE

Additional Editor Comments (optional):

Accept
---

## [Editor Report · Acceptance letter]

PONE-D-25-00479R2

PLOS ONE

Dear Dr. Ismail,

I'm pleased to inform you that your manuscript has been deemed suitable for publication in PLOS ONE. Congratulations! Your manuscript is now being handed over to our production team.

Kind regards,

on behalf of

Mr. Agbotiname Lucky Imoize

Academic Editor

PLOS ONE